# THINKING LIKE TRANSFORMERS

## ABSTRACT

What is the computational model behind a transformer? Where recurrent neural networks have direct parallels in finite state machines, allowing clear discussion and thought around architecture variants or trained models, transformers have no such familiar parallel. In this paper we aim to change that, proposing a *computational model for the transformer-encoder* in the form of a programming language. We map the basic components of a transformer-encoder – attention and feed-forward computation – into the simple primitives of `select`, `aggregate` and `zipmap`, around which we form a programming language: the *Restricted Access Sequence Processing Language (*RASP*)*. We show how RASP can be used to program solutions to tasks that could conceivably be learned by a transformer, augmenting it with tools we discover in our work. In particular, we provide RASP programs for histograms, sorting, and even logical inference similar to that of Clark et al. (2020). We further use our model to relate their difficulty in terms of the number of required layers and attention heads. Finally, we see how insights gained from our abstraction might be used to explain phenomena seen in recent works.

## 1 INTRODUCTION

While Yun et al. (2019) show that sufficiently large transformers can approximate any constant-length sequence-to-sequence function, and Hahn (2019) provides theoretical limitations on their ability to compute functions on unbounded input length, neither of these provide insight on *how* a transformer may achieve a specific task. Orthogonally, Bhattamishra et al. (2020) provide transformer constructions for several counting languages, but this also does not direct us towards a general model.

This is in stark contrast to other neural network architectures, which do have clear computational models. For example, convolution networks are seen as as a sequence of filters (Zhang et al., 2018), and finite-state automata and their variants have been extensively used both for extraction from and theoretical analysis of recurrent neural networks (RNNs) (Omlin & Giles, 1996; Weiss et al., 2018; Rabusseau et al., 2018; Merrill et al., 2020), even inspiring new RNN variants (Joulin & Mikolov, 2015).

In this work we propose a *computational model for the transformer-encoder*, in the form of a simple sequence-processing language which we dub RASP(*Restricted Access Sequence Processing Language*). Much like how automata describe the token-by-token processing behavior of an RNN, our language captures the unique information flow constraints under which a transformer (Vaswani et al., 2017) operates as it processes input sequences.

Considering computation problems and their implementation in the RASP language allows us to "think like a transformer" while abstracting away the technical details of a neural network in favor of symbolic programs. A RASP program operates on *sequences of values* from uniform atomic types, and transforms them by composing a restricted set of sequence processors. One pair of processors is used to *select* inputs for aggregation, and then *aggregate* the selected items. Another processor performs arbitrary but local computation over its (localized) input. However, access to the complete sequence is available *only* through aggregate operations that reduce a stream of numbers to a scalar. The key to performing complex global computations under this model is to compose the aggregations such that they gather the correct information, that can then be locally processed for a final output.

Given a RASP program, we can analyze it to infer the minimal number of layers and maximum number of heads that is required to implement it as a transformer. We show several examples of expressive programs written in the RASP language, showing how complex operations can be

```
1  def sort(vals,keys):
2    num_prevs = count_conditioned(
3                  (keys,indices),(keys,indices),
4                  lambda key,ind,key',ind':
5                      (key' < key) or
6                      (key' == key and ind' < ind))
7    select_sorted_val = select(indices,num_prevs,lambda i,np:i == np)
8    return aggregate(select_sorted_val,vals)
```

Figure 1: RASP program taking two sequences vals,keys and returning a sequence y sorting the elements of vals according to keys, e.g.: if vals($x$)=[a,b,c] and keys($x$)=[0,4,2], then y($x$)=[a,c,b].

implemented by a transformer. Thinking in terms of the RASP model also allows us to shed light on recent empirical observation of transformer variants (Press et al., 2020) and find concrete limitations of "efficient transformers" with restricted-attention (Tay et al., 2020).

## 2 THE RESTRICTED ACCESS SEQUENCE PROCESSING LANGUAGE

In this section, we present the *the Restricted Access Sequence Processing Language* (RASP). RASP assumes a machine composed of several Turing-complete processors, each of which can only run functions taking and returning a fixed number of primitive arguments, and a simple memory accessor that is controlled by these processors. The select, aggregate, and zipmap operations which we present will define and constrain how the processors work together to process an input sequence.

We will focus here only on the language itself, leaving the discussion of its exact relation to transformers to Section 3.

**Overview**  A RASP program works by manipulating *sequences*, occasionally with the help of *selectors*. Sequences contain values of uniform atomic type, such as booleans, integers, floats, or strings. They are functions used for selecting elements from sequences, and are used (together with the appropriate operations) only in the process of creating new sequences. All sequences in RASP are lazily evaluated, meaning that their length and contents are not populated until passed an input.

**The Base Sequences**  Every program in RASP begins from the same set of base sequences, and then creates new ones using a small number of core operations. These base sequences are **indices**, **length**, and **tokens**, evaluated on input $x_1, x_2, ..., x_n$ as their names suggest: $(0, 1, ..., n-1)$, $(n, n, ..., n)$ (of length $n$), and $(x_1, x_2, ..., x_n)$, respectively.

**Combining Sequences**  Sequences can be combined in an 'elementwise' manner, such that the value of the resulting sequence at each position $i$ is a function of the values in the combined sequences at position $i$ (similar to a map operation), or have positions 'mixed' in more complicated ways using *selectors*, which are functions $f : \mathbb{N} \times \mathbb{N} \to \{\text{True}, \text{False}\}$ whose sole purpose is to guide the combination of existing sequences into new ones.

We present the basic ingredients of RASP using an example. Figure 1 shows a simple RASP function for sorting a sequence of values according to a sequence of keys. It accepts an input sequence vals and uses the base sequence indices, that is available to any RASP program, to compute its output in three operations as follows:

1. count_conditioned of Line 2 creates a new sequence that counts for each element of keys the number of "previous items" it has in keys, where the "previous items" are defined to be all items that have a lesser value, or equal value and lower index. Thus, num_prevs creates a sequence of numbers, representing the target sorted position of each item.

2. select of line 7 creates a new *selector* which will focus each position $i$ on the corresponding position $j$ for which indices[$i$] is equal to num_prevs[$j$]. Effectively, it will direct the elements in each position $j$ towards their target location $i$.

3. Finally, `aggregate` of line 8 applies `select_sorted_val` to `vals`, moving each $i$-th element of `vals` to its calculated sorted position `num_prevs[`$i$`]`.

We now describe the base operations of RASP in-depth, occasionally presenting an example on the hypothetical input sequence $x$ of length $n$.

- **zipmap** The `zipmap` operation takes a tuple of sequences and an element-processing function `f`, and applies `f` per-index to the values in those sequences to create a new sequence. For a simple example, `y1=zipmap((indices,indices), lambda i,j:i+j)` creates a sequence that always evaluates to $(0, 2, ..., 2n - 2)$.

- **aggregate** The `aggregate` operation takes a selector `s`, a sequence `x`, and an optional parameter `default`, and averages subsets of the values of `x` into a new sequence `y` as follows: for every index $i$, `y[`$i$`]` is the average of `x[`$j_0$`]`, `x[`$j_1$`]`, ...`x[`$j_k$`]`, where $j_0, j_1, ..., j_k$ are the indices $j \in [n]$ for which `s(`$i,j$`)` is `True`. We say $k$ is the focus width of `s` at $i$. If $k = 0$, then `y[`$j_0$`]` is assigned the value in `default`. For example: if `s(`$i, j$`)` returns `True` iff $i$ is odd and $j$=0, and the value in default is `d`, then `y` will evaluate to `(d,x[0],d,x[0],...,y[`$n - 1$`])` where `y[`$n-1$`]` is either `d` or `x[0]` depending on the parity of $n$.

- **select** The `select` operation takes two sequences-tuples of lengths $k$ and $l$, `me=(`$m_1$`,`$m_2$`,...,`$m_k$`)` and `other=(`$ot_1$`,`$ot_2$`,...,`$ot_l$`)`, and a function `f` expecting $k + 1$ atomic values and giving boolean output. It composes these to create a selector `s` as follows: for every two indices $i, j$, `s(`$i, j$`)` is the output of `f` on the $i$-th and $j$-th slice of `me` and `other` respectively, i.e., `s(`$i,j$`)=f(`$m_1$`[`$i$`],...,`$m_k$`[`$i$`],`$ot_1$`[`$j$`]...`$ot_l$`[`$j$`])`. For a simple example, in `s=select((indices,),(indices,),lambda mi,oti:mi%2==1 and oti==0)`, then `m`$_1$`=indices`, `ot`$_1$`=indices`, and `s` is the same selector we used for our example in `aggregate` above.

- **count_conditioned** This operation takes the same parameters `me`,`other` and `f` as `select`, but this time returns a sequence `y` describing the *number* of selected influencing positions $j$ for each output position $i$ that `s=select(me,other,f)` would have created. In other words, for each $i$, `y[`$i$`]`$= k$ where $j_1, ..., j_k$ is the set of positions $j$ for which `s(`$i, j$`)=True`. For example, `h=count_conditioned((tokens,),(tokens,),lambda a,b:a==b)` returns an in-place histogram for the tokens in the input sequence: `h(`"abaa"`)`$=(3, 1, 3, 3)$.

This concludes the base operations of RASP – all other operations are shortcuts for combinations of the above 4, occasionally with the base sequences.

**Sugar** We implement RASP with a variety of syntactic sugar, presented fully in appendix E. Briefly:

1. When applying `zipmap` to a single sequence, it may be passed directly without using a tuple, e.g.: `zipmap(indices,f)` is equivalent to `zipmap((indices,),f)`.

2. `zipmap` has sugar for most of the binary operators, e.g.: for two sequences `x,y`, then `x+y` is sugar for `zipmap((x,y),lambda a,b:a+b)`.

3. Whenever the focus width of `s` at some index is $\leq 1$ ("up-to-one selection"), `aggregate(s,x,default=d)` does not explicitly compute the division. In this case the values of `x` do not have to be numbers.

4. `aggregate` accepts one additional optional parameter `elementwise_function`. The full order of parameters is `s,x,elementwise_function,default`, and the use of `elementwise_function` is as follows: `aggregate(s,x,f,d)` is equivalent to `aggregate(s,zipmap(x,f),default=d)`.

## 2.1 EXAMPLES

We now present some more example RASP programs, by increasing order of complexity.

***Simple Examples*** The first and simplest example is to compute an in-place histogram for some sequence `vals`. This is achieved with a single application of `count_conditioned`: `histogram=count_conditioned(vals,vals,lambda a,b:a==b)`.

```
1   def by_frequency(vals,default):
2     hist = count_conditioned(vals,vals,lambda a,b:a==b)
3     num_earlier = count_conditioned(
4                             (indices,vals),(indices,vals),
5                             lambda iq,vq,ik,vk:(vq==vk) and (ik<iq))
6     has_earlier = num_earlier > 0
7     unique_vals = zipmap((vals,has_earlier),
8                           lambda t,h_e: t if not h_e else default)
9     masked_hist = hist - (length * has_earlier)
10    return sort(unique_vals,key=-masked_hist)
```

Figure 2: RASP program sorting the unique elements of a sequence vals by decreasing frequency. Lines 6 and 7 show syntactic sugar for multiple simple zipmap operations, e.g., line 6 can be written has_earlier = zipmap(num_earlier,lambda n:n>0). This program requires a default atom da of the same type as vals, to put in place of all otherwise-duplicated values in its result. For example, by_frequency(tokens_str,"-")("abacca") will return "acb--".

***From Length to Parity*** While length is provided as a primitive in the language, it can actually be achieved as a composition of the other base operations and sequences. This is done by computing full_s=select((),(),lambda :True) followed by 1/aggregate(full_s,indices,lambda i:int(i==0)) (the fraction of elements equal to $0$ in indices, inverted). From length and that same full_s we can then define count(vals,v), a function taking any sequence vals and value v and returning a new sequence counting the number of appearances of v in vals. The implementation of count is simply length*aggregate(full_s,vals,lambda e:e==v). count in turn enables us to write programs like parity simply as count(tokens,1)%2==0[1].

***Reverse*** We can reverse a sequence seq with the help of an up-to-one selector mapping each position to its opposite: flip_s = select(indices, length-1-indices, lambda m,oth:m==oth). We use flip_s to re-order the tokens of seq: reverse=aggregate(flip_s,seq).

***Balanced Parentheses*** For balanced parentheses we use count_conditioned twice, storing in prev_opens and prev_closes the number of previous "(" or ")" (respectively) tokens each position has, including itself. The sequence is balanced if prev_opens-prev_closes has no negative values, and is $0$ at position length-1. These two qualities can be easily computed using two final select-aggregate pairs, and then combined with a zipmap.

***Most Frequent Tokens*** In fig. 2 we show how RASP can be used to arrange for any input sequence s the most frequent tokens in s, without repetition. The solution has 2 parts: first, we compute the histogram for all the tokens, and mask it such that all but the first of each token is given a negative value. Then, we sort the tokens according to the masked histogram. The solution uses the sort function from Figure 1.

```
1     def count_conditioned(me,other,f):
2       other = other+(indices,) # tuple concatenation, indices at end
3       s_with_0 = select(me,other,lambda *a:f(*(a[:-1])) or (a[-1]==0))
4       s_just_0 = select(me,other,lambda *a:f(*(a[:-1])) and (a[-1]==0))
5       frac = aggregate(s_with_0,indices,lambda i:int(i==0))
6       count_outside_0 = (1/frac)-1
7       count_in_0 = aggregate(s_with_0,(),lambda :1,default=0)
8       return count_outside_0 + count_in_0
```

Figure 3: Implementation of the operation count_conditioned in terms of indices and the other base operations, assuming that other is indeed a tuple of sequences (and not a single one).

---

[1]This does not contradict with the findings of Hahn (2019), who showed that parity is not computable in transformers *when each selector is restricted to width 1* ("hard attention").

***Count Conditioned*** The operation `count_conditioned` is a powerful part of RASP, appearing in many other programs. Surprisingly, it is realisable as a composition of the other operations (and base sequence `indices`). Understanding its implementation is interesting for learning how to "truly" think like a transformer, and we present the code in Figure 3. The intuition is as follows: we compute the select whose width we want to calculate twice, once such that it *also* selects the position $0$, and once such that it *only* selects this position. We then aggregate both these values, broadcasting $1$ from position $0$ and $0$ from everywhere else, and using default value $0$. The first `aggregate` computes for each position the *inverse* of the number of selected positions (excluding $0$) plus one, and the second computes whether that position would also focus on $0$. A straightforward `zipmaps` then gives us the result. To further help the intuition, we present in fig. 4 the computation flow for a `histogram` calculation, when `count_conditioned` is implemented as described here.

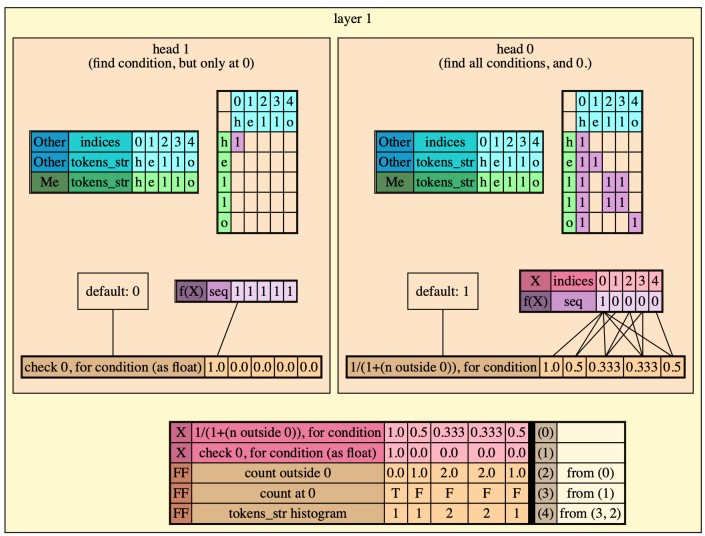

Figure 4: The computation flow for the sequence `histogram(tokens_str)` applied to input sequence "hello", when implemented in terms of `select`, `aggregate`, and `zipmap`.

***Note*** Many of the functions provided with RASP can be expressed in terms of `count_conditioned` – such as `count` (which counts how many elements in a sequence are equal to a given value) and `contains` (which checks if `count` is greater than $0$) – but this is not necessarily an optimal implementation with respect to number of heads it uses. The RASPlibrary provides optimal implementations.

## 2.2 IMPLEMENTING SYMBOLIC REASONING A LA ROVER

How might a transformer implement *reasoning*, as in Clark et al. (2020)? RASP empowers us to clearly think about such problems, and we sketch a solution direction here. We begin by reformulating the task of Clark et al. to a form that focuses on the core problem, moving away from natural language into more 'concrete' domain, and by limiting the type of relations we will consider.

In the work of Clark et al., a transformer is presented a sequence of statements and a query Q, and must recognise whether Q is implied by the previous relations or not. For example: a1∈A1, b∈A1 $\implies$ b∈A2, a1∈A2? evaluates to True, whereas a1∉A1, a1∉A2? evaluates to False. Different inputs for this task can have different *depth*: the number of 'logical hops' needed to correctly identify whether the query statement is true.

***Note.*** The original work accepts this input in natural language, e.g., "Alan is young. [...] If someone is young then...", and allows statements with more complicated logical form, such as a1∈A1∧a1∈A2 $\implies$ a1∈B. In this section we consider only a simplified and symbolic version, in which the statements are limited to the form of the previous paragraph. We assume the statements are separated by a special token `|`.

We sketch this task in RASP as follows: first, mark each statement 'block' in the sequence by its index (as a block), by counting for each token the number of separators before it in the input. Set aside the

final block (the query), it is the set of tokens with no separators after them. For each remaining block mark whether it is a relation ($\in$ or $\notin$) or inference ($\mapsto$) statement. Then, for each relation block, note at the position of the set token the element and whether it is inside or outside, and similarly over the element token note the set. Next, for as many repetitions as the logical depth that the program should cover: share set information between all elements and element information between all sets (including from inference blocks, which initially have none initially empty), and then apply one logical step 'locally' at each inference block. Finally, for the query, seek any occurrence of the set token in the sequence, and return whether the element token is listed there in its contents.

**Generalisation on Inference Depth**    A similar solution to the one we have proposed would be to make all of these logical inferences backwards from the query, i.e. by propagating backwards the requirements that would be sufficient to answer the query. If a trained transformer implements both of these solutions in parallel (for instance, to increase its robustness), this may explain the generalisation to greater query depth observed by Clark et al..

## 3    RELATION TO TRANSFORMERS, AND ANALYSIS

We discuss how RASP relates to the real transformer architecture, and how it may also be used to compare transformer variants, or analyse the 'difficulty' of a task for transformers.

**Connection of RASP to Transformers**    The `select` and `aggregate` operations of RASP correspond to the attention-scoring and then pooling of transformer self-attention heads, the `zipmap` operations correspond to the feed-forward sublayers, and the computed sequences correspond to head or feedforward inputs and outputs. `indices` and `tokens` represent the initial input, while `length` and `conditioned_contains` are in fact combinations of the other primitives. The persistence of sequences – such that they may be accessed multiple times over the course of a RASP program – is encouraged by the existence of skip connections in the transformer architecture. In appendix A we consolidate these connections, giving a full description of transformers, and showing how any given transformer can be represented in the RASP language (provided a slight generalization of `select`[2]).

### 3.1    PREDICTING TRANSFORMER COMPLEXITY WITH RASP

The purpose of RASP is to help us reason about the computation process enabled by a transformer. We find that RASP programs lend themselves easily to 'width' and 'depth' analysis, enabling us to predict the number of heads and layers that a transformer will need to implement the same solution. We discuss this analysis now, and evaluate the predictions it provides in Appendix B.

For any given RASP program, we can compute the minimal number of layers required to implement it in a transformer, and upper bound the number of heads this implementation requires.[3] This is provided its internal dimensions are wide enough to replicate the given processing functions. This analysis can give us intuition regarding the relative difficulty of different tasks for the transformer architecture, where each algorithm we find for a task gives us an upper bound on the number of layers and heads a transformer needs to solve it.

We implement such an analysis and provide a `draw_comp_flow` function, which automatically positions each attention head using a greedy scheduler, and displays the computation flow accordingly (see fig. 4, and others in the supplementary material).

A similar analysis can be done for different "primitive" computations in isolation, giving us intuition on the 'cost' of various common computations: how many additional heads and layers each computation adds when applied to a previously computed sequence. For example, our implementation of `sort` takes 2 layers, and so whenever we apply it to a computed sequence we know it will increase our program's depth by 2 from that sequence.

---

[2]This version is omitted from RASP purely in the interest of clarity for the programmer.

[3]The reason the number of heads can only be upper bounded is because it is possible that some `selects` in the program are equivalent, or can be combined into a single `select` without interfering with each other, but it is impossible to identify this statically.

**Algorithm for computing program depth and width**    As noted, the first 3 base operations of RASP – select, aggregate, and zipmap – have direct parallels in the transformer architecture, and so we may easily analyse the result of any RASP program to see how many layers and heads it would take to realise in an actual transformer. (For length and count_conditioned, we analyse them in terms of their deconstruction to the other operations and sequences.)

The first part of the analysis is simple: every sequence and selector is initiated with a "minimum depth" $d$, reflecting the earliest layer at which it can be computed, and so the minimum number of layers needed to create any given sequence is $d$. $d$ is computed for each new sequence as follows:

1. The base sequences indices and tokens have $d = 0$, as they are input to the transformer rather than part of its computation.

2. Any sequence created from a zipmap is given $d$ equal to the maximum $d$ of the inputs to that zipmap, as it can be created immediately after the last of them in the same feed forward computation that concludes it.[4]

3. Every selector gets $d$ equal to the maximum $d$ of its creating sequences X plus 1, to reflect the fact that all of them must be calculated before it can even begin (as multi-headed attention happens only once, at the beginning of each layer).

4. Any selector created from an aggregate has $d$ equal to at least that of its creating selector, and at least one plus those of the input sequences to the aggregate operation (as they must be passed through the attention to create the new sequences).

RASP makes it easy to access *all* sequence and selectors on which an unfinished value is dependent, and allowing us to analyse not only the depth but also the width of a given program. The *width* of the computation is a reflection of how many (unique) selectors are being used at every layer of the transformer: the number of attention heads needed to mimick that layer in a transformer. We say that a selector $s$ is being used at some layer $l$ if a sequence that is aggregated from $s$ is calculated at $l$. This is because it is possible a selector $s$ may have minimum depth $d$, but is only or also used at later layers: for instance if the sequences needed for the aggregation operation using $s$ are not yet ready.

Similarly, a sequence does not *need* to be computed at its minimum possible depth, as it is possible it will only be needed much later. Hence there is no one analysis for a given program, and there is room for creating a scheduling algorithm that minimises the maximum width of the transformer, i.e., the maximum of the widths of all layers (useful, as transformers tend to be created with uniform width).

## 4    IMPLICATIONS FOR TRANSFORMER VARIANTS

### 4.1    RESTRICTED-ATTENTION TRANSFORMERS

Multiple works propose restricting the attention mechanism of transformers in order to create more efficient transformers, reducing the time complexity of each layer from $O(n^2)$ to $O(nlog(n))$ or even $O(n)$ with respect to the input sequence length $n$ (see Tay et al. (2020) for a survey of such approaches and their complexity). Several of these do so using *sparse attention*, in which the attention is masked using different patterns to reduce the number of locations that can interact (see for instance (Child et al., 2019; Beltagy et al., 2020; Ainslie et al., 2020; Zaheer et al., 2020; Roy et al., 2020)).

Considering these variants of transformers in terms of the RASP language, allows us to reason about the computations they can and cannot perform. In terms of RASP, these variants of transformers all impose restrictions on the selectors, forcing some of the $n^2$ index pairs $(i, j)$ to False.

Figure 1 showed how to implement sorting of an input sequence with arbitrary alphabet size, comparison function, and length[5]. We now prove that RASP variants where the selector is restricted to $O(n)$ pairs (i.e., transformer variants with sufficiently restricted attention), cannot sort.

---

[4]Except in the special case where this maximum $d$ is 0, and the zipmap is not being called from within an aggregate, in which case the sequence is assigned $d = 1$. This reflects that at least one layer must be used to reach the first feed-forward computation, but also that attention may do a little processing itself before it "aggregates", using the linear translation **V**.

[5]Providing sufficiently stable word and positional embeddings, a practical limitation that applies to all transformer variants.

The computation model of RASP and indeed of all transformer variants allows comparison of values between more than one sequence position only during the `select` operation, i.e., only while computing the attention distribution Hence, all comparisons necessary for sorting must be applied in `select`. It follows that whenever `select` is restricted such that it compares at most $O(n)$ index pairs per head, no constant number of heads and layers will be sufficient for the model to perform sorting on arbitrary length – as sorting is known to require at least $O(n log(n))$ comparisons.

Thus, variants of transformers in which the attention is masked to impose $O(n)$ complexity require at least $O(log(n))$ layers to sort. It also follows that they require $O(log(n))$ layers to implement `count_conditioned`, as we see in fig. 1 that `count_conditioned` can be applied to create a sequence (`num_prevs`) which is sufficient to complete a sorting operation with only $O(n)$ further operations.

## 4.2 SANDWICH TRANSFORMERS

Recently, Press et al. (2020) showed that reordering the attention and feed-forward sublayers of a transformer affects its ability to train on language modeling tasks. In particular, they showed that 1. pushing feed-forward sublayers towards the bottom of a transformer weakened it, and 2. pushing attention sublayers to the bottom and feed-forward sublayers to the top strengthened it, provided there was still some interleaving in the middle (making a *sandwich transformer*).

Considering the base operations of RASP helps us understand the observations of Press et al.. In RASP, the feed-forward and attention sublayers are the `zipmap` and `select-aggregate` (or **gather** for short) operations. Any arrangement of the sublayers into a set architecture, from the 'vanilla' transformer to the variations considered in (Press et al., 2020), imposes a restriction on the number and order of RASP operations that can be chained in a RASP program. For example, an architecture in which all feed-forward sublayers appear before the attention sublayers imposes that no `zipmap` operation may be applied to the results of any `gather` operation.

In RASP, there is no value to repeated applications of `zipmap` before the first `gather`, as no further information can be generated beyond that already described by `indices` and `tokens`. This immediately explains the first observation of Press et al. (2020). Conversely, an architecture beginning with several attention sublayers – i.e., multiple `gather` operations – will be able to gather a large amount of information into each position early in the computation, if only by simple rules. More complicated gathering rules can be realised by applying `zipmap`s to the gathered information before generating new `selectors`[6], explaining the interleaved attention/feed-forward middle section present in the discovered architecture.

## 5 EXPERIMENTS

To evaluate the relevance of the RASP language to transformers in practice, we train transformers on a small set of synthetic tasks and compare their results to the head- and layer- bounds and attention patterns predicted by RASP programs for the same tasks.

While no RASP program promises to be a *unique* solution to the task it solves, several of the trained networks find solutions similar to those predicted by RASP. Among the most striking of these is the transformer trained to compute an in-place histogram, e.g. §abbd$\mapsto (1, 1, 2, 2, 1)$. We considered this task when the input sequences are presented with a beginning-of-sequence (BOS) token §, writing a single-head RASP program for it and training a single-head transformer on it. Visualizing the selection/attention patterns of these two heads (one RASP and one transformer) showed an identical pattern – see Figure 5.

We present the single-head RASP program for this task in Figure 6. Its operation is as follows: first, the selector `same_and_0` focuses each position $i$ on all positions $j$ containing the same token as $i$, and also on the position 0. Hence the width of this selector at each position $i \neq 0$ is exactly one plus the value $v_i$ that should be output at $i$. Aggregating the sequence $(1, 0, 0, ..., 0)$ with this selector (which always includes focus on 0) gives us the value $a_i = \frac{1}{v_i+1}$ at each location $i$, from which $v_i$ can then be recovered with a simple `zipmap`.

---

[6]Actually, the unbounded power of the processing functions `f` that RASP allows passing into `select` and `aggregate` technically renders `zipmap` unnecessary.

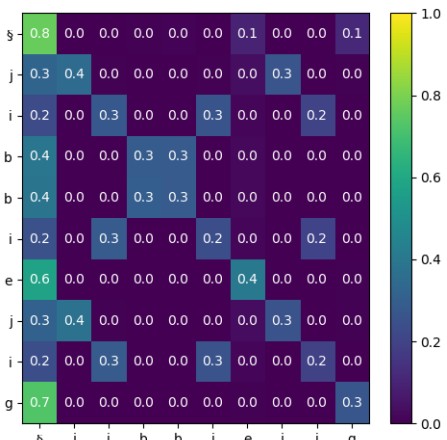 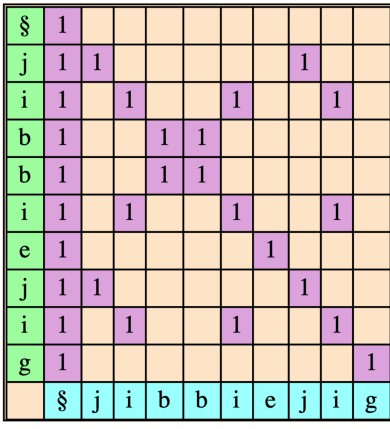

Figure 5: The attention (left) and selection (right) patterns of a single-head transformer and single-head RASP program both trained/written for the task of in-place histograms. We visualise both on the input sequence §jibbiejig. On the y axis, the sequence acts as the query, describing the output locations for which new values must be computed, and on the x axis it acts as the key, describing the input locations being selected to compute these new values. Specifically, each row in the attention pattern describes the self-attention distribution of this head over the sequence §jibbiejig. The transformer has clearly learned the same pattern as that used by the RASP program.

```
1    same_and_0 = select(tokens,(tokens,indices),
2                    lambda t1,t2,i:(t1==t2) or (i==0))
3    inverted_count = aggregate(same_and_0,indices,
4                            lambda i:int(i==0))
5    histogram_given_BOS = (1/inverted_count)-1
```

Figure 6: Building the RASP sequence `histogram_given_BOS` that computes an in-place histogram on the input tokens, under the assumption that the first token is always a special beginning-of-sequence character §. The selector `same_and_0` builds on the same intuition as the `s_with_0` selector from the implementation of `count_conditioned`.

The direct parallel between our program's only selector and our trained transformer's attention pattern (Figure 5) suggests that this RASP program describes the exact mechanism that our transformer has discovered.

We present further experiments on additional tasks in Appendix B.

## 6 CONCLUSIONS

We abstract the computation model of the Transformer-encoder into a simple sequence processing language that captures the constraints on information flow in a Transformer. Considering computation problems and their implementation in the RASP language allows us to "think like a transformer" while abstracting away the technical details of a neural network in favor of symbolic programs. Moreover, provided it uses reasonable element-processing functions, we can analyze any RASP program to infer the minimum number of layers and maximum number of heads required to implement it in a transformer. We show several examples of expressive programs written in the RASP language, showing how complex operations can be implemented by a transformer. We train several transformers on these tasks, and find that RASP helps us predict both the correct number of heads and layers to use for these tasks and also the attention patterns that the transformers realise to solve them. Additionally, we use RASP to shed light on an empirical observation over transformer variants made by Press et al. (2020), and find concrete limitations of some "efficient transformers" architectures.

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

## A    TRANSFORMERS IN RASP

RASP is almost – but not quite – a strict over-approximation of transformer-encoders. In this section, we show how the addition of `score`, a generalisation of `select` that may assign non-boolean values to index pairs, makes RASP a strict over-approximation. In particular, we give an explicit translation from any given transformer-encoder to a RASP program, provided the augmentation with `score`. If the reader prefers to start there, a full description of transformers is given in section D (with notations in the preceding section).

We now introduce the `score` operation, and expand `aggregate` to receive a `scorer`:

- **score** Similarly to `select`, the `score` operation takes two tuples of sequences `me=(m1,...,mk)` and `other=(ot1,...,otl)`, and a function `f` expecting k+l atomic values. This time however, `f` may return any non-negative float value. It creates from these a `scorer` similarly to how `select` creates a `selector` from its inputs.

- **aggregate** The `aggregate` operation is expanded such that it may receive either a `scorer` or a `selector` where it previously accepted only a `selector`. When it receives a `scorer`, each `y[i]` is assigned the weighted average of all the values of `x`, according to the values in the `scorer`:

$$y[i] = \frac{\sum_{j \in [n]} s(i,j) \cdot x[j]}{\sum_{j \in [n]} s(i,j)}$$

Intuitively, the `select` operation, and the application of `aggregate` to a `selector`, can be seen as the special case of a `score-select` pair in which the `scorer` has only given scores of $0$ and $1$. It does have one difference in that it also allows for a default value which may be used when the total score for some index `i` is $0$. However, this can be seen as sugar: it is not difficult to create a mechanism similar to that of `count_conditioned` in order to recognize when a selector has width $0$, and so avoid the direct use of `default` in aggregate[7]. Moreover, if the data is always given with a special beginning-of-sequence (BOS) token, then the default value can simply be loaded from that location whenever no other focus is found.

---

[7]The head and layer analysis of RASP can be trivially updated to take this into account.

```
1   def elementwise_part(x,att_y,el_funcs):
2     lA, n2, ff = el_funcs
3     def apply_elementwise_parts(xy):
4       xy = tovec(xy)
5       x, att_y = first_half(xy), second_half(xy)
6       x1 = x + lA(att_y) # vector addition
7       res = x1 + ff(n2(x1))
8       return totuple(res)
9     return zipmap(x+att_y,apply_elementwise_parts)
10
11
12  def layer(x,head_weights,n1,el_funcs):
13    multihead_y = ()
14    for lq,lk,lv in head_weights:
15      s = score(x,x,lambda xixj:
16          exp(scalarprod(
17                lq(n1(first_half(xixj))),
18                lk(n1(second_half(xixj))))))
19      head_y = aggregate(s,x,lv)
20      multihead_y = multihead_y + head_y
21      # + here is concatenating tuples of sequences
22    return elementwise_part(x,multihead_y,el_funcs)
23
24  def T_y0(layer_funcs,w,p):
25    x = zipmap((indices,tokens),lambda i,t:p(i)+w(t))
26    for head_weights,n1,el_funcs in layer_funcs:
27      x = layer(x,head_weights,n1,el_funcs)
28    return x # tuple of d float sequences
```

Figure 7: Function $\mathsf{T\_y0}$ that takes a description of a transformer and embedding function and returns $d$ RASP+score sequences mimicking the output values of that transformer exactly (where $d$ is the output dimension of that transformer). Note that the loops in $\mathsf{T\_y0}$ and $\mathsf{layer}$ are 'meta-loops' with respect to the generated RASP program: they generate multiple layers and heads, but the number of these is a function of the transformer being mimicked and not of the input it will receive.

**Theorem A.1.** *Let $\mathcal{T} : (\mathbb{R}^d)^* \to (\mathbb{R}^d)^*$ be a transformer and $y_0 : \Sigma \to (\mathbb{R}^d)^*$ be an input embedding computed as the sum of a token and positional embedding, $y_0(x)_i = w(x_i) + p(i)$. Then the computation of $\mathcal{T}_{y_0} \triangleq y_0 \circ \mathcal{T}$ can be mimicked in a RASP program that writes the output to $d$ float-sequences and uses exactly $LH$ score and aggregate calls and $(H+1)L+1$ zipmap calls, where $L$ and $H$ are the number of layers and attention heads in $\mathcal{T}$, respectively.*

***Proof Sketch*** In figure 7 we present code that, given the token and positional embedding $w : \Sigma \to \mathbb{R}^d$ and $p : \mathbb{N} \to \mathbb{R}^d$ and all the weights of a transformer, recreates that same transformer in RASP. Our code relies on the helper functions tup2vec, vec2tup, first_half and second_half which help convert between the tuples of values given to the processing functions by zipmap and aggregate and the vectors they represent. For simplicity in the presentation, we assume here that the transformer is given as a collection of linear transformations, layer-norms, and feed-forward functions which can be applied to its internal vectors directly.

The main routine is $\mathsf{T\_y0}$, which applies the initial embedding and stores it in a tuple of $d$ sequences, x, and then applies each layer to it in turn. This takes $L$ calls to the layer function, each of which computes score–aggregate (with additional zipmap before the aggregate) pair $H$ times to mimic each of the heads, and then calls a zipmap on the concatenation of their result to complete the remaining (elementwise) computations of the layer.

***Note.*** Recall that, as noted in E, when a processing function passed to zipmap returns multiple values, zipmap simply generates that same number of sequences. In particular, at all iterations of the loop in $\mathsf{T\_y0}$, x is a tuple of $d$ sequences where $d$ is the embedding dimension of the given transformer.

We see that, when augmented with score, the RASP language naturally composes the components of a given transformer to reconstruct it exactly.

**Why not have score?**   The motivation for the omission of score from RASP is cleanliness: it is far easier to think in terms of select as opposed to score, and we have not yet encountered a problem where we used scorers whose values where outside of $0$ and $1$. In time, as we use the language more and encounter the limitations this choice poses, we may return to score and see what other kind of special cases of it we will benefit from including in RASP.

**Power of RASP**   As seen in this section, RASP can (provided this slight generalization of select) represent any transformer. Additionally, we see that it is not arbitrarily overpowered. For example, it does not allow iterating over a sequence of arbitrary length one-by-one to perform some gradual computation (as might be done in RNN or DFA), and in general does not allow arbitrary repetition of operations as other languages might allow. This is because the number of operations in a RASP program is predetermined: RASP programs do not admit loops. This distinction between transformers and RNNs is known, and there is interest in bridging it. For example, the Universal Transformer attempts to introduce loops into transformers, by allowing them to also have a control mechanism that decides whether to repeat a layer during computation  (Dehghani et al., 2018).

## B   EXPERIMENTS

For RASP to be useful, it is important to see that RASP programs relate well to actual transformer behavior in practice. In this section, we train transformers on a small set of synthetic tasks for which we can write RASP programs, and see how the these programs relate to our empirical results.

We consider the following tasks:

1. **Count-a**: Return the number of 'a' tokens in a sequence, e.g., aba$\mapsto 2$.
2. **Histogram**: For each token, its number of repetitions in a sequence, presented in-place. For example, aababc$\mapsto (3, 3, 2, 3, 2, 1)$.
3. **Reverse**: Reversing a sequence, e.g.: abc$\mapsto$cba.

For Histogram, we also consider a variant with a special beginning-of-sequence (BOS) token §, appearing exactly once at the beginning of each sequence (and nowhere else). For example, §aabc$\mapsto (1, 2, 2, 1, 1)$.

We train transformers for each of these tasks, and test whether our RASP programs accurately predict the minimum number of heads and layers needed to perform them. We also visualise their attention distributions[8], and see if they match the selectors used by our programs.

We find that several of the RASP programs presented in this paper show similar attention (selector) patterns to those of the trained transformers in practice, suggesting that programming in RASP helps us provide reasonable predictions of transformer behavior. Moreover, we often find that reducing the number of heads and layers in a transformer beyond the number needed in our RASP program for the same task significantly degrades its accuracy. This suggests that the specific programs we have presented in for these tasks are also optimal solutions.

**Data and Evaluation**   Unless stated otherwise: for all of the languages, we use the alphabet {a,b,c,d,e} with sequences of sizes 1 through 100. These are generated by first choosing the length uniformly from 1 to 100, and then choosing each token uniformly from the alphabet. We use $50,000$ train samples, $1,000$ test samples, and $1,000$ validation samples.

For tasks giving a 'single' output value – such as Count-a (which gives 1 number) as opposed to Reverse which gives a new sequence – we train the network to return that value in all positions, e.g., aba$\mapsto (2, 2, 2)$ for Count-a. This makes the visualisation of the attention distributions clearer (as all locations are trying to do something meaningful, as opposed to one), and is also more clearly aligned with the tasks we have described in this work.

We measure the accuracy of a transformer on a batch of sequences as the fraction of total predictions it made for those sequences that were correct, e.g. $\frac{x}{5}$ for a batch with total sequence length 5. For the train, set, and validation sets, we report accuracy as the average batch accuracy.

---

[8]i.e., after softmaxing the attention scores.

**Architecture** We use the transformer architecture provided with PyTorch 1.7.0, with an additional single linear transformation and softmax at the end to convert to the output classes prediction. Unless stated otherwise, we use small embedding and feed-forward dimensions: 20 and 40, respectively. We vary the number of heads and layers per task.

**Training Method** We train with the ADAM optimizer, sin-cosine positional embedding, and dropout 0.1 in the transformer weights. We use learning rate 0.0003, batch size 50, and PyTorch's `ExponentialLR` learning rate scheduler with gamma 0.98 (updated after every epoch). Excluding confirmation of a negative result, we train each network for 20 epochs. If a network hits 100% accuracy before then, we stop.

**Helpful RASP functions** In this section we will make frequent use of the full-selector `full_s=select((),(),lambda :True)` and the function `frac_condition(sequences,f)`, which computes `aggregate(full_s,sequences,lambda a:int(f(*a)))`[9] – the fraction of input positions for which the `f` is satisfied on the values of `sequences`. Note that `frac_condition` requires only 1 layer (after `sequences` have been computed) with 1 head, and that that head is `full_s`. Recall also that `length` is computed: `length=1/frac_condition(indices,lambda i:i==0)`.

## B.1 COUNT-a

To avoid gradient problems from trying to obtain large numerical values from the transformer (e.g., 8), we encode Count-a as a categorical task. In particular, we create 21 output tokens {0,1,2,...,20}, and if there are more than 20 `a` tokens in the sequence we just report 20.



Figure 8: The attention distribution in the only head of a transformer trained on Count-a. The sequence is presented on the y-axis as the queries and on the x-axis as the keys, i.e., each row is a distribution over the input positions. As predicted by our RASP program (which solves this task using `full_s`), this distribution is relatively uniform.

RASP permits a 1-layer, 1-head program for this task: `count_a = lengthfrac_condition( tokens, lambda t:t=="a")`. (`length` and `frac_condition` share the selector `full_s`.) Accordingly, training a transformer with 1 layer and 1 head on Count-a succeeded, reaching test accuracy 98.9% on the 20th epoch.

The attention pattern of this transformer on the input sequence abbaabcde is shown in Figure 8. While the distribution is not perfectly uniform, it does seem to focus on the entire sequence, as the use of `full_s` in our RASP program suggests (contrast for example with the attention distribution for Reverse, in Figure 12)[10].

---

[9]The exact implementation is slightly different to account for the case when there is only one `sequence`, but the idea is the same.

[10](We leave here a cautionary tale: if you do not properly scale the colorbar of your attention figure, it will look like the difference between focus on different locations is much greater than it is!)

We stress that this is a different distribution to that which we might intuitively expect for this task – namely, an attention pattern focused solely on instances of a in the sequence – and that RASP has successfully pushed us to predict it!

## B.2 HISTOGRAM

As with Count-a, we encode histograms as a categorical task. This time we limit the maximum count to 10, i.e., we only use the output tokens 1,2,...,10. (0 is irrelevant as it will not appear in an in-place histogram). Unlike most other tasks, for Histograms we use an alphabet of size 10: {a,b,...,j}. For histograms, we use an input alphabet of size 10: {a,b,...,j}.

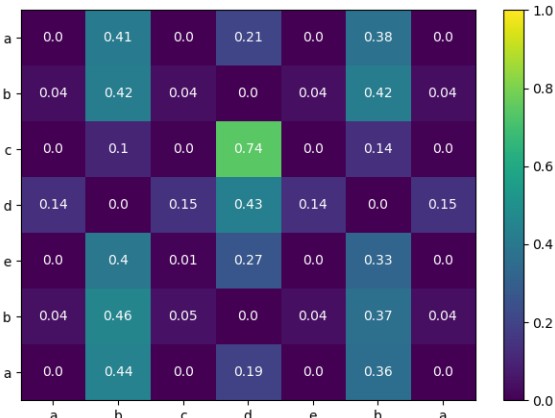

Figure 9: The attention distribution in the single head transformer trained on Histogram. The input sequence abcdeba is presented on the y-axis as the queries and on the x-axis as the keys, i.e., each row is a distribution over the input positions. This transformer does not have enough heads: despite training for 50 epochs, it has only reached test accuracy 55%, and generates the incorrect output $(1, 2, 1, 1, 1, 2, 1)$ for this sequence. Similarly, it does not manage to recreate the selection pattern predicted by our RASP program for this task, which requires 2 heads.

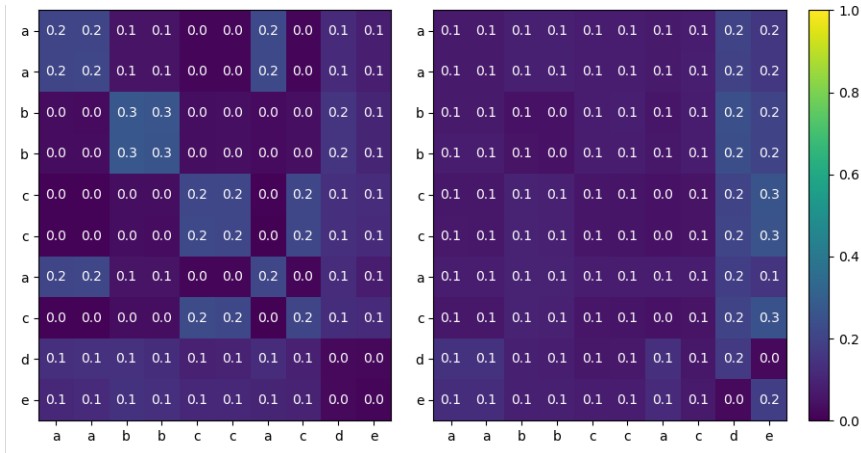

Figure 10: The attention distribution in the 2-head, single-layer transformer trained on Histogram. The input sequence aabbccacde is presented on the y-axis as the queries and on the x-axis as the keys, i.e., each row is a distribution over the input positions. The distribution is similar, but not identical, to our count_conditioned implementation. In particular, it seems that the tokens d and e are to some extent taking the role of the index 0 in our implementation: the first head has most tokens focus on themselves, d, and e, and the second head has most tokens focus slightly more on d and e than others. d and e themselves seem to behave inversely to the other tokens.

We trained one transformer with 1 layer and 2 heads, and another with only 1 layer and 1 head. After 20 epochs, the transformer with 2 heads reached test accuracy 89.3%. In contrast, after 50 epochs, the transformer with only 1 head was still at accuracy 55%! Increasing its embedding and feed-forward dimensions to 50 and 100 respectively also did not work: a 1-layer 1-head transformer with these dimensions still only reached 79.3% test accuracy after 50 epochs, and this after being past 77+% validation accuracy since the 27th epoch.

Drawing the attention map for the single-head transformer (Figure 9) did not seem to relate to the selection pattern of count_conditioned at all, unsurprisingly considering that it did not have enough heads. See for example the apparent focus on b by several query positions not containing b, as opposed to our expectations of the count_conditioned focus pattern as shown in Figure 4.

For the 2-heads transformer, we draw its attention maps in Figure 10. The solution has some clear parallels with our RASP prediction of histogram selection patterns of count_conditioned, such as most tokens focusing on themselves in the one head and sharing distribution patterns in the other. But we can also easily find differences: in particular, the d and e tokens seem to avoid rather than focus on themselves, and the shared focus of most tokens in the similar-attentions head is not on $0$ but on d and e. There is a possibility that in this transformer, d and e are playing a role similar to that which we gave to $0$ in our implementation of count_conditioned. We leave a deeper exploration of this to future work.

### B.3    HISTOGRAM WITH BOS

Recall the implementation of count_conditioned (Figure 3): it calculates the "width" (number of selected locations) of a hypothetical selector s by simulating it along two actual selectors, s_with_0 and s_just_0. s_with_0 is used to calculate for every index the fraction $1/c'_i+1$ where $c'_i$ is the width of s on everything except index $0$, and s_just_0 is used to make a final adjustment from $c'_i$ to the actual width, depending on the focus on $0$.

If, then, the contents at position $0$ are constant across all inputs, then the second selector s_just_0 becomes unnecessary: any information it conveys can be hard-coded into the RASP program (practically, the transformer). It follows that for setups where all input sequences are prepended with a special beginning-of-sequence (BOS) token, count_conditioned can be implemented with only one head, using just the s_with_0 selection pattern.

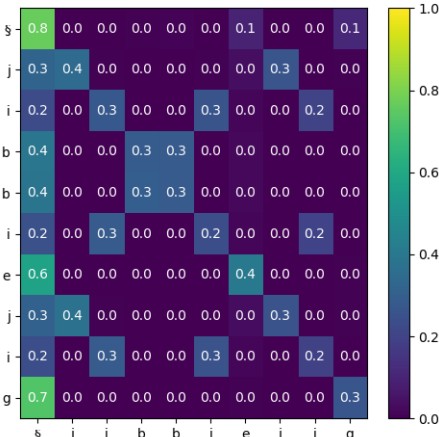

Figure 11: The attention distribution for a 1-head, 1-layer transformer trained on Histogram with BOS tokens. The input sequence jibbiejig is presented on the y-axis as the queries and on the x-axis as the keys, i.e., each row is a distribution over the input positions. This transformer appears to have implemented Histogram (with BOS) using exactly the same s_with_0 selection pattern as suggested in our count_conditioned implementation! (The second selection pattern in the count_conditioned implementation, s_just_0, is unnecessary when the value at the $0$ index is constant.))

We prepend all of the original Histogram inputs with a special BOS token § (and their outputs with 1), and train a new 1-layer, 1-head transformer on the resulting data set. For gamma=0.99, the results satisfy our predictions perfectly: the transformer reaches 99.7% test accuracy in 20 epochs, and drawing its attention distribution (Figure 11) shows that it follows exactly the pattern of the selector `s_with_0`! (For gamma=0.98 the attention distribution was also very similar to that of `s_with_0`, but the model reached only 86.4% test accuracy after 20 epochs.)

**Discussion of BOS** The significance of such 'non-input' tokens in transformers has been previously discussed, with different interpretations. For example, Vig & Belinkov (2019) refer to the attention focused on the initial token of a sequence – seemingly when there is nothing else to focus on – as *null attention*. They report that the null token gathered as much as 97% of the attention of some of their heads, and suggest that this is consistent with these heads being unimportant to the transformer's overall performance. Conversely, this new result suggests that the null token at the beginning of a sequence is playing an important role in the transformer calculations, and in particular is directly assisting in counting!

### B.4 REVERSE

In RASP, Reverse is implemented using `flip_s=select(indices,length-1-indices,lambda i,j:i==j)` followed by `reverse=aggregate(flip_s,tokens)`. This takes two layers of one head each (recall that `length` itself requires one layer to compute).

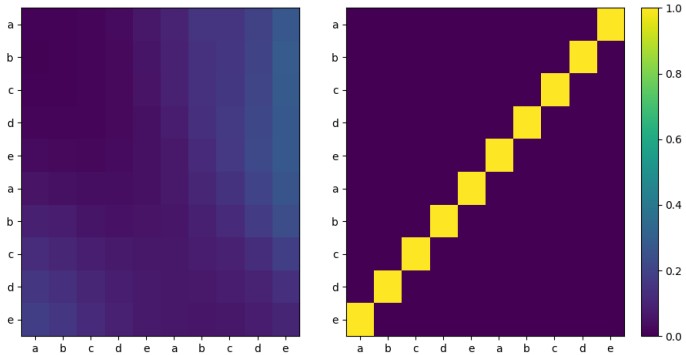

Figure 12: The attention distributions in the 2-layer, 1-head (per layer) transformer trained on Reverse. The first layer is on the left and the second on the right. The input sequence abcdeabcde is presented on the y-axis as the queries and on the x-axis as the keys, i.e., each row is a distribution over the input positions. The attention distribution in the second layer is exactly as predicted by our RASP program (i.e., a hard, 'reverse-match' attention). In the first layer however, it seems the transformer has learned a behavior other than uniform attention to compute the length of the sequence (which it needs for the second layer).

We train two transformers on Reverse: one with 2 layers and 1 head, and the other with 1 layer and 2 heads, to verify that the separation into 2 layers is indeed necessary. To give room for the index-based selection pattern (i.e., a scoring method that involves comparison of indices and not just tokens), we give the transformers per-head width at least as large as our maximum length. In particular, we use embedding dimension 100 for the 2-layer transformer and 200 for the 1-layer transformer[11]. We also give them each feed-forward dimension twice their embedding dimension.

The 2-layer transformer reaches test accuracy 99.6% after 20 epochs. In contrast, and as expected, the single-layer transformer remains trapped at 39.6% test accuracy even after 50 epochs. Plotting the attention for the 2-layer transformer (Figure 12) matches some of the predictions of our RASP program: the reverse-matching attention is only computed in the 2nd layer, and computed perfectly at that point. We are inclined to believe the length is being computed at the first layer (as we predict).

---

[11]Intuitively, this allows the query and key vectors to encode their positions/target sources as one-hot vectors, matching each other perfectly when computing the attention scores. We leave a full exploration of the relation between embedding dimension and selector complexity to later work.

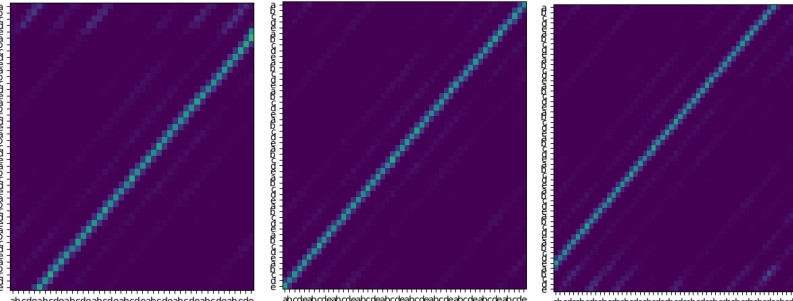

Figure 13: The attention distributions in the 1-layer, 1-head transformer trained on Reverse with fixed input length 50, on input sequences of length 45, 50, and 55, in order. The y-axes represent output locations (queries) and the x-axes input locations (keys), i.e., each row is a distribution over the input positions. As expected, the transformer has learned a constant relation between pairs of input locations, and maintains all of the pairs it can find in each input sequence it gets. For missing pairs, such as $(0, 49)$ through $(4, 45)$ in the input of length 45 (left), it struggles to focus its attention.

But there are also devitions from our prediction: the attention pattern suggests that the transformer is computing the length using a different mechanism from the one that we have suggested.

We now strengthen the claim that the initial layer of the Reverse transformer is computing the sequence length, by showing that when the sequence length is fixed, then a single-layer (and single-head) transformer *does* succeed on Reverse. We fix the sequence length to 50 and train a 1-layer 1-head transformer on Reverse. This simpler task can be presented in one layer and one head in RASP using the single selector `flip50_s=select(indices,49-indices,lambda i,j:i==j)`, from which the result is computed `reverse50=aggregate(flip50_s,tokens)`.

As expected, the transformer succeeds in its task, reaching $100\%$ test accuracy in only 3 epochs. In Figure 13 we illustrate that it has indeed learned a constant $(i, 49 - i)$ location pairing, by visualising its attention on sequences slightly longer or shorter than it has been trained on.

For completeness, in Figure 14 we show also the attention patterns of the single-layer transformer trained on variable-length Reverse. In keeping with our predictions, it has not managed to learn the reverse-matching attention pattern at all. This is because it needs an additional layer to compute the length before it can create the correct attention pattern.

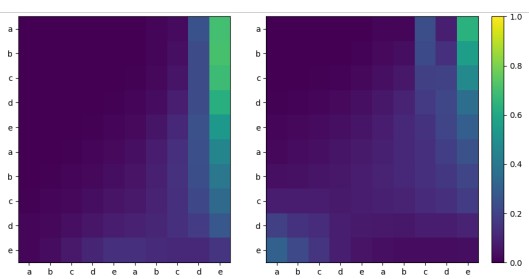

Figure 14: The attention distributions of the 2 heads in the 1-layer transformer trained on Reverse, as applied to the sequence abcdeabcde. As expected from the RASP program, the transformer is unable to learn the reverse-match attention on any head of its first layer.

## C   NOTATIONS

**Basic Notations**     For every $n \in \mathbb{N}$, we denote $[n] = \{1, ..., n\}$.

**Matrices** For a matrix $X \in \mathbb{R}^{n \times d}$, we refer to its $i$-th row as $X_i \in \mathbb{R}^d$, and for a vector $v \in \mathbb{R}^d$ we refer to its $i$-th value as $v_i \in \mathbb{R}$. We additionally refer to $n$ as the length of $X$ and abusively denote $|X| = n$.

For $X \in \mathbb{R}^{n \times d}$ and $b \in \mathbb{R}^d$, we use the shorthand $X + b$ to describe the addition of $b$ to each of the rows of $X$, i.e.: $(X + b)_i = X_i + b$ for every $i \in [n]$. For a scalar $\alpha \in \mathbb{R}$, any operation $X \odot \alpha, \odot \in \{+, -, \times, \div\}$ is applied elementwise to all the values of $X$. Matrix multiplication between two matrices $A, B$ is denoted simply $AB$, and the transpose of a matrix $A$ is denoted $A^T$.

We occasionally treat input or output sequences $x_1, ..., x_n \in \mathbb{R}^d$ as matrices $X \in \mathbb{R}^{n \times d}$ whose rows are the individual input vectors: $X_i = x_i$. When $n$ may be arbitrary, we will say that $X \in (\mathbb{R}^d)^*$.

**Definition C.1.** *A* linear transformation *with input dimension $d$ and output dimension $m$ is a function* $l_{A,b} : (\mathbb{R}^d)^* \to (\mathbb{R}^m)^*$ *parameterised by a matrix $A \in \mathbb{R}^{d \times m}$ and vector $b \in \mathbb{R}^m$ as follows:* $l_{A,b}(X) = XA + b$ *for every $X \in (\mathbb{R}^d)^*$. $A$ and $b$ are the* weights *of the transformation.*

**Definition C.2.** *The* softmax *function* $\mathrm{s} : \mathbb{R}^* \to \mathbb{R}^*$ *is defined: for every $d \in \mathbb{N}$ and $x \in \mathbb{R}^d$,* $\mathrm{s}(x) \in \mathbb{R}^d$ *such that* $\mathrm{s}(x)_i = \frac{e^{x_i}}{\sum_{j \in [d]} e^{x_j}}$ *for every $i \in [d]$. We also denote by $\mathcal{S}$ the row-wise softmax function: for every $n, d \in \mathbb{N}$ and $X \in \mathbb{R}^{n \times d}$, $\mathcal{S}(X) \in \mathbb{R}^{n \times d}$, and $\mathcal{S}(X)_i = \mathrm{s}(X_i)$ for every $i \in [n]$.*

We denote by $\mathcal{R}(X)$ the elementwise application of the ReLU function, $r : x \mapsto \max(0, x)$, to $X$.

**Function Qualities** A function $f : A^* \to B^*$ is *length preserving* if it satisfies $|f(x)| = |x|$ for any $x \in (\mathbb{R}^d)^*$ (i.e., for any input sequence $x_1, ..., x_n \in A$, $f$ returns a sequence $y_1, ..., y_n \in B$). If there also exists a function $g : A \to B$ such that $f(x)_i = g(x_i)$ for any $i \leq |x|$, then $f$ is *elementwise*, and we say that $f$ is an *elementwise application* of $g$. Note that linear transformations are elementwise.

# D TRANSFORMER-ENCODERS

At the highest level, a transformer-encoder $T$ (henceforth, a *transformer*) is a parameterised length-preserving function $T : (\mathbb{R}^d)^* \to (\mathbb{R}^d)^*$ composed of multiple layers of length-preserving functions $\ell : (\mathbb{R}^d)^* \to (\mathbb{R}^d)^*$, i.e. $f = \ell_1 \circ \ell_2 ... \circ \ell_L$ which we will describe in this section.

Generally speaking, a transformer's layers are not elementwise, and indeed the transformer would not be interesting if they were. However, when we come to look at their components, we see that this quality rests entirely only on their use of *attention*[12].

**Attention** Attention is a function devised to enable 'recollection' of previously processed data from a history of arbitrary length (Bahdanau et al., 2015; Luong et al., 2015). Transformers use a variant called *scaled dot-product attention* to collect data from multiple locations in an input sequence.

**Definition D.1.** Scaled Dot-Product Attention *is a function* $\mathrm{a} : (\mathbb{R}^d)^* \to (\mathbb{R}^m)^*$ *parameterised by 3 linear transformations, $l_Q, l_K, l_V : (\mathbb{R}^d)^* \to (\mathbb{R}^m)^*$ and defined for every $X \in (\mathbb{R}^d)^*$ as follows:*

$$\mathrm{a}(X) = \mathcal{S}\left(\frac{l_Q(X) l_K(X)^T}{\sqrt{m}}\right) l_V(X)$$

***Note.*** The original definition of scaled dot-product attention allows $l_Q, l_K$, and $l_V$ to have different output dimensions $m$. In this case, the denominator is $\sqrt{m_k}$.

For convenience, from here on we refer to scaled dot-product attention simply as *attention*.

The attention computation can be broken into 3 stages. First, a pairwise score is calculated for each pair of locations, showing how much the input in location $j$ should influence the output in location $i$: this is the value $\mathbb{S}_{i,j}$ in the matrix $\mathbb{S} = \frac{l_Q(X) l_K(X)^T}{\sqrt{m}}$. Then, each input is processed in-place ($l_V(X)$) to create candidate outputs, and finally the candidate outputs are averaged for each output location $i$, according to the softmaxed scores $\mathcal{S}(\mathbb{S})_i$ for that location. In this sense, attention can be seen as a request to gather into each output information from various locations, where $l_Q$ and $l_K$ work together to select information sources, and $l_V$ encodes the transferred information.

---

[12]Unsurprisingly, given the title of the paper.

Transformer layers often gather information with multiple attention functions, referred to as *attention heads*, whose results are concatenated back into a single output at the end:

**Definition D.2.** *Let $d, H, m \in \mathbb{N}$ be such that $d = Hm$. A* multi-headed attention *function with input dimension $d$ and $H$ heads is a function $\mathcal{A} : (\mathbb{R}^d)^* \to (\mathbb{R}^d)^*$ parameterised by the weights of $H$ scaled dot-product attention functions $\mathrm{a}_1, ..., \mathrm{a}_H$ as follows:* [13] *for every $X \in (\mathbb{R}^d)^*$,*

$$\mathcal{A}(X) = \mathrm{a}_1(X) \cdot \mathrm{a}_2(X) \cdot ... \cdot \mathrm{a}_H(X)$$

*where $\cdot$ denotes row-wise concatenation, i.e. for every $i \in [|X|]$, $\mathcal{A}(X)_i$ is the concatenation of $\mathrm{a}_1(X)_i$ through $\mathrm{a}_H(X)_i$. The functions $\mathrm{a}_1, ..., \mathrm{a}_H$ are referred to as the* heads *of $\mathcal{A}$.*

In addition to attention, transformers use *layer-norm* and *feed-forward* components, as follows:

**Definition D.3.** *A* single-row layer-norm *over dimension $d$ is a function $g : \mathbb{R}^d \to \mathbb{R}^d$ parameterised by vectors $a, b \in \mathbb{R}^d$ and constant $\varepsilon \in \mathbb{R}$, and defined for every $x \in \mathbb{R}^d$ and $i \leq d$ as follows:*

$$g(x)_i = \frac{a_i(x_i - \bar{x})}{\mathrm{std}(x) + \varepsilon} + b_i$$

*where $\bar{x} = \frac{\sum_{j \in [d]} x_j}{d}$ is the mean of $x$ and $\mathrm{std}(x) = \sqrt{\frac{1}{d-1} \sum_{i \leq d} (x_i - \bar{x})^2}$ is its standard deviation.*

*A* layer-norm *over dimension $d$, $\mathbf{n} : (\mathbb{R}^d)^* \to (\mathbb{R}^d)^*$, is an elementwise application of a single-row layer-norm of dimension $d$.*

The layer-norm's function is more a reguliser, and indeed, it will not play a part in our abstraction.

**Definition D.4.** *A* feed-forward *function with input dimension $d$ and internal dimension $m$ is an elementwise function $\mathcal{F} : (\mathbb{R}^d)^* \to (\mathbb{R}^d)^*$ obtained by composing two linear transformations $L_1 : (\mathbb{R}^d)^* \to (\mathbb{R}^m)^*, L_2 : (\mathbb{R}^m)^* \to (\mathbb{R}^d)^*$ and ReLU, as follows:*[14] *$\mathcal{F}(X) \triangleq L_2(\mathcal{R}(L_1(X)))$.*

The feed-forward component is elementwise, and the combination of two linear transformations with nonlinear activation provides strong expressive capacity (Hornik et al., 1989).

**Definition D.5** (Transformer-Encoder Layer)**.** *A transformer-encoder layer with input dimension $d$, internal dimension $m$, and $H$ heads (such that $d/H \in \mathbb{N}$), is a length-preserving function $\ell : (\mathbb{R}^d)^* \to (\mathbb{R}^d)^*$ composed of one multi-headed attention $\mathcal{A}$ with input dimension $d$ and $H$ heads, one feed forward function $\mathcal{F}$ with input dimension $d$ and internal dimension $m$, two layer-norm functions $\mathbf{n}_1, \mathbf{n}_2$ over $d$, and one linear transformation $l_A : (\mathbb{R}^d)^* \to (\mathbb{R}^d)^*$, as follows: for every $X \in (\mathbb{R}^d)^*$,*

$$X_1 = X + l_A(\mathcal{A}(\mathbf{n}_1(X))) \tag{1}$$
$$\ell(X) = X_1 + \mathcal{F}(\mathbf{n}_2(X_1)) \tag{2}$$

We refer to the additions in both equations as a 'skip connection'. Note that the layernorm, feed-forward, and skip connection components of the layer are elementwise: were it not for the attention, the entire layer would be elementwise.

Finally, we present the full encoder architecture:

**Definition D.6** (Transformer-Encoder Vaswani et al. (2017))**.** *A transformer-encoder with $L$ layers, $H$ heads, and input and internal dimensions $d, m$ is a length-preserving function $\mathcal{T} : (\mathbb{R}^d)^* \to (\mathbb{R}^d)^*$ parameterised by the weights of $L$ transformer-encoder layers $\ell_1, ..., \ell_L$, each with $H$ heads and input and internal dimensions $d, m$, and defined for every $X \in (\mathbb{R}^d)^*$ as follows:*

$$\mathcal{T}(X) = \ell_L(...\ell_2(\ell_1(X)))$$

**Permutation Invariance of Transformers**   An interesting trait of the transformer architecture is that it has no inherent positional awareness. Specifically: for any transformer $\mathcal{T} : (\mathbb{R}^d)^* \to (\mathbb{R}^d)^*$, input sequence $\mathbf{x} = x_1, x_2, ..., x_n \in \mathbb{R}^d$, and permutation $\pi$, we have $\mathcal{T}(\pi(\mathbf{x})) = \pi(\mathcal{T}(\mathbf{x}))$ [15].

---

[13]Some definitions of multi-headed attention may present it with an additional '$+X$' in the computation (representing a 'skip connection' present in the transformer), or final linear transformation applied to the result. For reasons that will be clarified later, we prefer to set the boundaries of the definition only to the direct 'mixing' operation shown, and instead write the skip connection and further linear transformation explicitly when presenting the transformer.

[14]During training, a dropout layer is also applied after the ReLU operation, but this is not present in inference.

[15]As all components of transformers other than attention are elementwise, we need only consider attention in order to be convinced of this. We see quickly that at each output location $i$, attention is a function only of

**Discrete Input** Transformers $\mathcal{T} : (\mathbb{R}^d)^* \to (\mathbb{R}^d)^*$ are used to process non-empty sequences over a finite alphabet $\Sigma$ by composing them with a simple length-preserving function, $y_0 : \Sigma^+ \to (\mathbb{R}^d)^*$: $\mathcal{T}_{y_0}(x) \triangleq \mathcal{T}(y_0(x))$. This $y_0$ is in turn composed from a token embedding $w : \Sigma \to \mathbb{R}^d$ and position embedding $p : \mathbb{N} \to \mathbb{R}^d$, which are normally combined using addition: for every $x = x_1...x_n \in \Sigma^*$, $y_0(x_1, x_2, ..., x_n)_i = w(x_i) + p(i)$[16].

From here, whenever we refer to a 'transformer over (some finite alphabet) $\Sigma$', we mean a transformer paired with an initial embedding $y_0$ as described above.

# E  ADDITIONAL DETAILS ABOUT RASP

**A note about types in RASP** Technically, there is no one 'tokens', rather 5 options: `tokens_str`, `tokens_int`, `tokens_float` and `tokens_bool` cast the input sequence to the corresponding atomic types, and `tokens_asis` takes the input sequence as-is. For brevity, we refer to all of these as `tokens` here.

**RASP Sequences** RASP operates exclusively on sequence- and selector-generating functions, which we refer to as *sequences* and *selectors* respectively and RASP-*functions* together. All RASP-functions take as input exactly one non-empty sequence, and when describing them and how RASP manipulates them to create new RASP-functions we will do so in terms of the sequences and selectors that they and their manipulations generate from each input sequence. When it is clear from context, we will simply refer to them as sequences and selectors, and describe them directly in terms of their outputs. For example, if we say that a RASP operation applies +1 elementwise to each value in a sequence u, we actually mean that it returns a new sequence v such that for every input $x$ and position $0 \le i < |x|$, `v(x)[i]=u(x)[i]+1`.

*Note* In this section we will refer to the $i$-th value in a sequence s as `s[`$i$`]`, such that `s[`$i$`]` is in one of the atomic types. We stress however that this is only for the discussion, and not a part of the language.

**Additional operations** For brevity in code, the RASP also comes with the following syntactic sugar:

- Anywhere that a tuple of sequences is passed into an operation, a single sequence may be passed in as-is as well. For example, `y=zipmap((indices,),(indices,),lambda a,b:a+b)` can also be written `y=zipmap(indices,indices,lambda a,b:a+b)`.

- The `zipmap` operation is accessible through a large range of operators, covering its application to all of the base binary and unary operations on the atomic types. For example, the above y can equivalently be defined as `y=indices+indices`. These operators can also be mixed with constants from the atomic primitives, such that an equal y be obtained using `y=2*indices`.

- `aggregate` may receive a tuple of sequences xx instead of a single sequence x. In this case, it returns a new tuple (of the same length) of sequences, the result of applying `aggregate` to each of the sequences in xx. For example, the line `a,b = aggregate(s,(x,y))` is equivalent to `a,b = aggregate(s,x), aggregate(s,y)`.

- The processing function passed to `zipmap` may return more than one value (provided the number of values it returns is constant). In this case, the operation will arrange the output values into the same number of output sequences. For example, `y1,y2=zipmap(x,lambda v:v+1,v+2)` is equivalent to writing `y1=x+1` and then `y2=x+2`.

- Anywhere that a processing function is expected, if one is not provided, the identity function is used[17].

---

$l_K(X), l_V(X)$, and $l_Q(X)_i$, where the order of the rows of $l_K(X)$ and $l_V(X)$ does not matter as long as they remain aligned.

[16]Note that without the position embedding, $y_0$ would be elementwise, and so its combination with $\mathcal{T}$ would be permutation invariant – an undesirable trait for sequence processing.

[17]This is consistent with `aggregate` allowing you to choose whether to pass a single sequence, or a tuple of sequences and a processing function.

- The function **select1** which takes only a single tuple of sequences `xx` and an index-computing function `fi`, and is syntactic sugar for `select(xx,(indices,),lambda *a:fi(a[:-1])==a[-1])`. `select1` is guaranteed to return a true select satisfying the "up-to-one" property, i.e., that can be successfully paired in aggregates with an `x` that does not contain numbers.

