# OpenReview forum: "Thinking Like Transformers"
_ICLR.cc/2021/Conference — Reject_

### Official Review · AnonReviewer3 · 2020-10-14
**Very interesting but underdeveloped**

**Rating:** 4
**Confidence:** 3

**Review:**

The authors introduce a DSL, the Restricted Access Sequence Processing (RASP) language, that they claim can serve as a computational model for the transformer-encoder.  They develop the reader's intuition for RASP by providing RASP implementations of many basic operations such as computing histograms, sorting, and reversing.  They also show how, for a given RASP program, to determine the minimum number of layers required and to upper-bound the number of heads required to implement it as a transformer.  Lastly, they analyze two transformer variants, restricted-attention transformers and sandwich transformers.  For the former, they use the RASP perspective to claim a theoretical limitation, and for the latter, they comment that a known empirical finding is intuitive in light of the RASP perspective.

I found this paper very interesting, a rare conceptual gem in a mostly empirical field.  The ideas in the paper open up many new questions and directions.  I wish I could champion it but sadly I find it critically underdeveloped in its current state and not yet ready for publication.

The main weakness of the current version is that it only glosses over the connection between RASP and the transformer-encoder.  It does not explain what it means to be a computational model for it, and does not provide general principles for abstracting DSLs from network architectures nor discuss the possible design space.  Section 3 says "While we give a detailed explanation [of the relationship between RASP and transformers] in the appendix..." but the appendix contains no such analysis, only more details of RASP in isolation.  I also find it strange and somewhat of a red-flag that RASP does not seem to be compositional, for example, it does not seem to include a computational model of feed-forward networks within it.  The `zipmap` operation nominally corresponds to the feed-forward stage of the transformer-encoder, but RASP does not seem to include any restrictions on the function being zip-mapped.

I think the paper could be strengthened significantly if it began with a DSL for building computation graphs sufficient to express transformers, and then presented a compositional (if not fully principled) way of abstracting this DSL into a traditional DSL that can serve as a reference computational model for the original version.  Ideally this process generalizes the two prior computational models they refer to, i.e. CNNs as sequences of filters and RNNs as state machines.  I think it is also critical to make explicit what properties are being relaxed and what differences are being abstracted away.  For example, would the authors consider RNNs and LSTMs to have the same reference computational model or would differences be preserved in the abstraction? Are there RASP programs that cannot be realized by transformers, or vice-versa? More generally, what does it mean for one language to be a computational model of another one, and what does the design space look like?

If the main weakness is that the authors do not ground their computational model to transformers theoretically, a related weakness is that they do not ground their computational model to transformers empirically either.  I think the paper could also be strengthened significantly by simple empirical experiments, for example using their RASP implementations of various simple functions to make predictions about the accuracy of transformers trained on those tasks as a function of the number of layers and heads they are provided.  I would be particularly interested to see how sharp these curves are. Is there a phase transition once the minimum required layers/heads are provided, or is it much more gradual?  Is there any evidence that the transformer actually learns these reference programs?

Miscellaneous comments:
- I found the description of RASP unnecessarily difficult to follow. There are conventions for introducing DSLs, e.g. presenting the grammar and then the semantics.  There are also some inconsistencies, with 'aggregate' introduced as taking two arguments but then used in the prose taking a mysterious third lambda argument.
- It is not immediately obvious what it means for one RASP function to call another as a subroutine.  Is it assumed that any such subroutine has one distinguished input that must always have the same size as the distinguished input to the original RASP function (so that `indices` and `length` are the same)?
- I think the inclusion of non-float types merits more discussion. For example, is it important that there is a boolean type?
- The section on logic programming is too informal, with phrases like "We suggest approaching this task in RASP as follows..." and "If a trained transformer ..., this may explain ...".  Does RASP permit one or more decision procedures for horn clauses? If so, what is the code?  Do you hypothesize that the transformers in (Clark et al. 2020) are learning hybrid forwards/backwards reasoning? If so, how might you test this hypothesis?
- The impossibility result for sorting should discuss non-comparison-based sorting algorithms (e.g. radix sort) or else qualify the claim.
- The RASP analysis of the sandwich transformer results (S4.2) does not seem particularly illuminating.
- There is essentially no discussion of related work.

Minor:
- the second sentence of the intro repeats "language"
- in 'The Base Sequences' paragraph, there is a double comma and a dangling close-paren
- the `sort` code uses `seq` instead of `vals` in the last line
- in explanation of `select`, `f` is referred to as a "selection function" even though it has a different type
- top of page 4: attention distribution <MISSING PERIOD> Hence

---

> ### Author Response · Authors · 2020-11-17
> **Response - Reviewer #3 - Part 1 of 3**
>
> Thank you for your detailed review, we appreciate your interest.
>
> **1. The main weakness of the current version is that it only glosses over the connection between RASP and the transformer-encoder. And later: Are there RASP programs that cannot be realized by transformers, or vice-versa? More generally, what does it mean for one language to be a computational model of another one, and what does the design space look like?**
>
> We apologise for this missing discussion! We are returning it to the paper. In particular, we are adding a discussion of these relations to the paper, as well as an appendix fully describing transformers and how they can be encoded in a slightly extended version of RASP.
>
> Briefly, the relation between the RASP operations and the transformer-encoder is as follows (this is a similar description to that given to reviewer 1):
>
> `indices` and `tokens` reflect the original input embedding applied to any transformer input. `select` reflects the computation of the attention distribution (in which the score between each two input locations is computed pairwise as a function of the embedded contents at those two locations alone), `aggregate` reflects the completion of the attention computation (having computed the distribution), and `zipmap` represents all of the local operations computed between each two attention sublayers (i.e., the feedforward, layernorm, and skip-connection operations).
>
> This results in an overall "information flow" (how the input tokens interact) identical to that of a transformer. For example, we cannot iterate over every item in the sequence, building information in some state updated one token at a time. We also cannot process the entire sequence together in order to choose the attention distribution (the “selector” in RASP). In general: we are restricted to sharing information between locations in the same way as a transformer.
>
> *RASP->transformer, transformer->RASP:*
> In particular, the operations are parallel enough that:
> (1) if we generalise slightly the `select` and `aggregate` operations, it is possible to present any transformer as a RASP program with identical number of heads and layers (we are adding a full description of this in the appendix), and conversely,
> (2): if we sufficiently restrict the element-processing functions (the inputs to `zipmap` and `select`) to properly reflect the feedforward and attention layers they represent, it would be possible to build a compiler from any RASP program to a transformer.
>
> *Design Space*
> The restriction along the lines of information flow is necessary for gaining insight into behavior of a transformer. For example, it is through this type of restriction that we realise that counting the number of occurrences of a token in a sequence is not done, as one might intuitively expect, by focusing the attention directly on those occurrences!
>
> Hence we argue that any language attempting to abstract transformers must also introduce operations restricting this flow, which would result in a series of operations similar to `zipmap`, `select`, and `aggregate`, right down to the pairwise behavior of the `select` computation (i.e., the relation `s(i,j)` for each two locations `i`,`j` is a function of the values in those two locations alone, and in particular not influenced by the other locations).
>
> The remaining degrees of freedom are in the types of functions that `zipmap` and `aggregate` may receive, and we note that RASP is fully compatible with accepting appropriate restrictions for these in the future.
>
> *Other NN architectures, and a "global" language:*
> To your comment on RNNs and their variants, we note that the information flow in these architectures is completely different to that of transformers: where RNNs maintain a fixed-size state that is updated exactly once for each token in the input sequence, in order, (i.e., a variable number of times), transformers maintain a "variable-size" (as a function of input length) state that is only updated a fixed number of times, but at each update can read the entire input sequence -- albeit in a carefully restricted way (through the attention).
> Similarly, the information flow in CNNs is different from that of transformers: there is no global attention mechanism.
> That said, it would be interesting in the future to develop a full language with different operations that compile to either transformers or RNNs, such that a RASP program may compile to a network consisting of multiple interleaving transformer and RNN layers!

---

> > ### Author Response · Authors · 2020-11-17
> > **Response - Reviewer #3 - Part 2 of 3**
> >
> > **2. RASP does not seem to be compositional, and later: It is not immediately obvious what it means for one RASP function to call another as a subroutine**
> >
> > RASP programs are lazily evaluated: `indices`, `tokens`, and `length` are in fact functions returning sequences, and not sequences in themselves. For example, for the base function `length`, `length(“abc”)` will return the sequence `[3,3,3]` and `length(“a”)` will return `[1]`. Hence the size of the sequences is determined always only on evaluation, and programs can be composed without explicitly sharing parameters to maintain a consistent size.
> >
> > For a simple example of RASP functions calling each other, consider the following RASP program, that checks for each token in a sequence whether it appears at least as c times in that sequence:
> >
> > ```
> > def input_histogram():
> > 	return count_conditioned(tokens,tokens,lambda a,b:a==b)
> >
> > def appears_c_times(c):
> > 	return input_histogram() > c
> > ```
> >
> > an example use of this program is as follows:
> > ```
> > >r3 = appears_c_times(3)
> > >r3(“abaabc”)
> > Sequence[True, False, True, True, False, False]
> > >r3(“aba”)
> > Sequence[False, False, False]
> > ```
> > To clarify: the functions `input_histogram` and `appears_c_times` do not return evaluated sequences, they return functions which generate sequences. These sequence-generating functions can then be re-used in other more complicated RASP programs.
> >
> > In particular, all RASP operations (`zipmap`, `select`, `aggregate`) do not apply directly to sequences, rather, they compose sequence-generating functions. `indices`, `tokens`, and `length` are the "global", base, functions of every RASP program.
> >
> > **3. RASP … does not seem to include a computational model of feed-forward networks within it. The `zipmap` operation nominally corresponds to the feed-forward stage of the transformer-encoder, but RASP does not seem to include any restrictions on the function being zip-mapped.**
> >
> > In this paper we focus more on the information flow during a transformer's processing of a sequence (i.e., information flow between the embeddings at each position and layer), than we do on the possible manipulations that can be done locally on this information. Hence for now we have not restricted the possible behaviors of the `zipmap` and `selection` functions that RASP accepts from the user, and it is the user’s responsibility not to abuse this power. As the goal of RASP is to help guide the user to think like a transformer, we think that this is reasonable.
> >
> > However, we agree that adding such restrictions is important, and intend in the future to define which operations are allowed for each of these functions. i.e., in the future, we expect to augment the language such that `zipmap` and `select` may not receive an arbitrary lambda but rather demand a special feed-forward or dot-product-attention abstraction. We believe that this will also allow us to more tightly bound what may or may not be expressed by a transformer. In turn we hope this will inspire seeking better attention mechanisms, as the tighter abstraction shines light on what we would like to express, but cannot.
> >
> > **4. I think the paper could also be strengthened significantly by simple empirical experiments, for example using their RASP implementations of various simple functions to make predictions about the accuracy of transformers trained on those tasks as a function of the number of layers and heads they are provided.**
> >
> > We agree and will be working on providing experimental results soon.
> >
> >
> > **5. I found the description of RASP unnecessarily difficult to follow. There are conventions for introducing DSLs, e.g. presenting the grammar and then the semantics.**
> >
> > We apologise, and will work on clarifying the presentation.
> >
> > **6. There are ... some inconsistencies, with `aggregate` introduced as taking two arguments but then used in the prose taking a mysterious third lambda argument.**
> >
> > This is some sugar whose introduction fell in editing, we will return it to the paper: `aggregate(s,x,f)` is equivalent to `aggregate(s,zipmap(x,f))`. This is also why you may see aggregate with a tuple of sequences given for `x` as opposed to a single sequence as expected.
> >
> > **7. I think the inclusion of non-float types merits more discussion. For example, is it important that there is a boolean type?**
> >
> > The value of these types is in reflecting the meaning we give to the sequences we are manipulating. RASP would be just as expressive without the boolean type, but given that the purpose of this language is in helping guide our thoughts about transformers, it would be less useful for its goal.
> >
> > **8. The section on logic programming is too informal. ... Does RASP permit one or more decision procedures for horn clauses? If so, what is the code?**
> >
> >
> > We will try to convert this section into pseudocode, which we hope will be more informative, and update here with it soon. If your question on horn clauses is separate from this, could you please clarify?

---

> > > ### Author Response · Authors · 2020-11-17
> > > **Response - Reviewer #3 - Part 3 of 3**
> > >
> > > **9. Do you hypothesize that the transformers in (Clark et al. 2020) are learning hybrid forwards/backwards reasoning? If so, how might you test this hypothesis?**
> > >
> > > Hybrid forwards/backwards reasoning would be one potential explanation for the generalisation seen in the work of Clark et al, and it would be in line with the possible solution we present here. One way to test this would be to take the model of Clark et al. that is trained for depth 4, freeze its lower layers, and train simple (e.g. one or two layer FF) classifiers on the outputs of these for different forwards and backwards inference depths (e.g. number logical hops from assumptions, and number of logical hops from conclusion when assuming conclusion is true but assumptions are not available). If we find that the classifiers succeed for “low forwards” and “low backwards” depths but not for “high forwards” and “high backwards” depths, we can conclude that it has learned both forwards and backwards reasoning. However, we leave testing this hypothesis fully to future work.
> > >
> > > As a concrete example to clarify the above: consider the input sequence: `[a in A], [A subset B], [B subset C], [C subset D], [D subset E], [a in E?]`. A forward-hop of depth 1 would allow us to conclude that `a in B`, but not that `a in C`, and similarly a backward hop of depth 1 would tell us that `a in D` is sufficient for our question but would not know that `a in C` is too. If applying a simple classifier to the initial layers of a transformer trained on such queries manages to learn `a in B` and `a in D?` but not `a in C` or `a in C?`, we might conclude that the transformer is propagating the information/query both forwards and backwards through the input clauses.
> > >
> > > **10. The impossibility result for sorting should discuss non-comparison-based sorting algorithms (e.g. radix sort) or else qualify the claim.**
> > >
> > > The significance of the claim is not so much in saying that the efficient transformers cannot sort, but in providing a formal proof that the count_conditioned operator simply cannot be realised by efficient transformers. In particular, the proof shows that an implementation of `count_conditioned` in a (linearly) efficient transformer implies a general sort in less than $O(nlog(n))$ time, without assumptions on the input data, which we know to be impossible. In other words, this result shows that the attention component of the vanilla transformer is truly “taking advantage” of at least $nlog(n)$ of the $n^2$ operations it takes to compute it!
> > >
> > > The question of whether or not efficient transformers can still implement non-comparison based sorting algorithms is independent of this claim.
> > >
> > >
> > > **11. There is essentially no discussion of related work.**
> > >
> > > You are correct, we will expand our discussion of relevant and motivating works in the paper. We note however that there is no room for a dedicated related work section: this is the first attempt we know of to create a programming language abstracting any neural network architecture, and so there is no existing work to compare to.
> > >
> > > **12. The sort code uses seq instead of vals in the last line**
> > >
> > > Thank you!
> > >
> > > **13. In explanation of select, `f` is referred to as a "selection function" even though it has a different type.**
> > >
> > > Thank you. We will update our terminology to be clearer. In particular, we will refer to all functions that operate directly on atomic types (i.e., the functions that `select` and `zipmap` accept as inputs) as ‘element functions’. This is to differentiate them from the sequences and selectors themselves, or the actual RASP operations (`zipmap`, `select`, and `aggregate`).

---

> ### Author Response · Authors · 2020-11-24
> **Thanks, and pointer to relevant changes in revision**
>
> Thank you again for your encouraging words and thoughtful comments.
>
> We recently uploaded a revision, and would like to point out major changes that may interest you:
> 1. We have added an appendix (A) discussing more in depth the connection between transformers and RASP. In particular, we show how a slight generalization of the `select` operation allows encoding any transformer in a RASP program with the same number of heads and layers.
> 2. We have added an experiments section, testing the relation between RASP programs and actual learned patterns in transformers. The full experiments section is in appendix B, and we give a brief recount in section 5 of the main paper. In figure 5 in the main paper we show how a transformer trained to compute histograms has done so by implementing the selection pattern of the RASP program for the same problem almost perfectly (showing the transformer pattern side-by-side with the RASP selection pattern). In the appendix we see for example how RASP correctly predicts that reversing a sequence requires two layers, and moreover strongly corresponds with the learned attention patterns.
>
> We have also edited the paper with respect to clarity and references to related work, for which we thank the helpful comments of all the reviewers.

---

### Official Review · AnonReviewer2 · 2020-10-21
**Interesting concept badly presented and evaluated**

**Rating:** 5
**Confidence:** 4

**Review:**

This paper proposes a restricted programming language containing rough analogues of operations used in transformers. Using this language, the authors show how some algorithms can be implemented, which gives some insights about the limitations of transformers.

Overall, the paper is badly written, with many typos and many hard-to-understand areas. Since this is a "thought-experiment" paper, this alone is a good reason for rejecting this work at its current form.

Some things I would have expected an analysis on:
* How well do the individual RASP operations map to the relevant transformer operations?
* Why are the examples in 2.1 important/useful? Where are they needed?
* What are "useful" algorithms that cannot be represented in RASP? How would we need to change a transformer to allow it to represent such algorithms?
*   Sec 2.2 is a textual description of a complicated algorithm, why not build it gradually from primitives by introducing larger functions? Experimental results on RASP/Transformers?
* Can you "compile" RASP into a transformer (=architecture+weights) that performs exactly the task defined in RASP? (If not, why? if yes, some experimental validation would be useful)



##### Typos
* Fig1, Line 8: should the second arg be `vals`?
* Sec2: "the base sequences": unbalanced parenthesis
* In the discussion of `aggregate` a selector `s` is an input, but `s` is never used. Should `s` be `f` instead ?
* In the definition of `select`, `s(i,j)=f(m1[i],...,mk[i], ot1[j], ... otl[j])` What does `m1[i]` mean? `m1` is the first element of `me` but `m1` is also a sequence somehow?
* Footnote 1 says uses variable `n`. What is `n`? Should it be `max(k,l)`?
* Fig3 the semantics of the operation in L2 have not been defined, similarly for Fig2 L7.

---

> ### Author Response · Authors · 2020-11-17
> **Response - Reviewer #2 - Part 1 of 2**
>
> Thank you for your detailed review.
>
> We apologise for the confusing presentation, and are working on clarifying the paper. We will upload a revised version (including incorporating other comments) in the coming days.
>
> **1. How well do the individual RASP operations map to the relevant transformer operations?**
>
> The relation between the RASP operations and the transformer is as follows: `indices` and `tokens` reflect the original input embedding applied to any transformer input. `select` reflects the computation of the attention distribution (in which the score between each two input locations is computed pairwise as a function of the embedded contents at those two locations alone), `aggregate` reflects the completion of the attention computation (having computed the distribution), and `zipmap` represents all of the local operations computed between each two attention sublayers (i.e., the feedforward, layernorm, and skip-connection operations). The operations are parallel enough that, if we generalise the `select` and `aggregate` operations, it is possible to present any transformer as a RASP program with identical number of heads and layers.
>
> In the opposite direction, RASP currently admits arbitrarily powered element-processing functions in the `zipmap` and `select` operations, and so in the general sense it is not possible to convert a given RASP program to a transformer. Hence, if a programmer wishes to write a RASP program that can be converted to a transformer, they must be careful not to abuse the power of these functions. As the goal of the language is to help in reasoning about transformers, we feel this is a reasonable expectation.
>
> In the future, we may develop languages that reflect specifically the feed-forward and attention-score operations. In this case, we may restrict RASP to only accept them in the `zipmap` and `select` operations, and any RASP program under this restriction would be directly compilable to a transformer without manual intervention from the programmer!
>
> For clarity, we are adding a discussion of these relations to the paper, as well as an appendix fully describing transformers and how they can be encoded in a slightly extended version of RASP.
>
>
> **2. Why are the examples in 2.1 important/useful? Where are they needed?**
>
> The point of the examples in 2.1 is to familiarize the reader with RASP, which we feel is important in a paper presenting a new language. It also helps show how expressive RASP is, which might not be immediately obvious from the base primitives - consider for example the solution for balanced parentheses (dyck-1), for which you might be initially inclined to imagine a "sequential" solution with the help of a counter!
>
> **3. What are "useful" algorithms that cannot be represented in RASP? How would we need to change a transformer to allow it to represent such algorithms?**
>
> Any task that "requires" a for-loop over sequence length - i.e. only has solutions that process the input sequence one-by-one over its tokens - would not be expressible in RASP (as the number of 'heads' and 'layers' in a RASP program is independent of its input, much like those of a transformer). The most relevant transformer variant for such algorithms would be the universal transformer (Dehghani et al., 2018), which may choose the number of times it repeats layers.
>
> **4. Sec 2.2 is a textual description of a complicated algorithm, why not build it gradually from primitives by introducing larger functions?**
>
> That is a good idea for clarifying sec 2.2, and we will do that.

---

> > ### Author Response · Authors · 2020-11-17
> > **Response - Reviewer #2 - Part 2 of 2**
> >
> > **5. Can you "compile" RASP into a transformer (=architecture+weights) that performs exactly the task defined in RASP?**
> >
> > Regarding 'compilation', we have not yet implemented a general compilation from RASP programs into a transformer (this is certainly a long term goal, and we will significantly restrict the zipmap and selection lambdas as we approach it). For now we note that RASP programs are  helpful for building intuition about transformers. Consider for example the recent paper of Bhattamishra et al ("On the ability of self attention networks to recognise counter languages"), in which they construct a transformer for recognising shuffle-dyck languages. If we study their work, we see that their solution may in fact be described by the following RASP program:
> >
> > ```
> > def contains_condition(mv,f):
> > 	condition_flag = zipmap(mv,f)
> > 	return count(condition_flag,True)>0
> >
> > def constant_mv(val):
> > 	return zipmap((),lambda t:val)
> >
> > def shuffle_k(paran_pairs):
> > 	select_prevs = select(indices,indices,lambda i,j:j<i)
> > 	select_last = select((),(indices,length),lambda i,n:i==n-1)
> > 	all_info = []
> > 	for opener,closer in paran_pairs: # eg "(" and ")"
> > 		running_total = aggregate(select_prevs,tokens,
> > 								lambda t: 1 if t==opener else -1)
> > 		has_negative_balance = contains_condition(running_total,
> > 							   	lambda rt:rt<0)
> > 		balanced_at_end = aggregate(select_last,running_total) == 0
> > 		all_info.append((has_negative_balance,balanced_at_end))
> > 	is_ok = constant_mv(True)
> > 	for has_negative_balance,balanced_at_end in all_info:
> > 		is_ok = is_ok and (not has_negative_balance) and balanced_at_end
> > 	return is_ok
> > ```
> >
> > an example use of this program is as follows:
> > ```
> > > dyck2 = shuffle_k([“()”,”[]”]
> > > dyck2(“([)]”)
> > Sequence[True,True,True,True]
> > ```
> >
> > Hence, while we may not have an algorithm for compiling RASP programs to transformers at this stage, we still believe they are helpful for reasoning about transformers.
> >
> > **6. Fig1, Line 8: should the second arg be vals?**
> >
> > Yes, thank you!
> >
> > **7. In the discussion of `aggregate` a selector `s` is an input, but `s` is never used. Should `s` be `f` instead ?**
> >
> > Yes, `f` and `s` refer to the same thing, thank you!
> >
> > **8. In the definition of `select`, `s(i,j)=f(m1[i],...,mk[i], ot1[j], ... otl[j])`. What does `m1[i]` mean?**
> >
> > `me` and `other` are each tuples of sequences, i.e., `m1...mk` and `ot1...otl` are all sequences. For example, in the example in the same paragraph, `m1` and `ot1` are both the sequence `indices`. For another example, in `select_last=select((),(indices,length),lambda i,n:i==(n-1))`, `me` is the empty tuple `()` and `other` is the tuple `(indices,length)`.
> >
> > **9. Footnote 1 says uses variable n. What is n? Should it be max(k,l)?**
> >
> > n is the length of the input sequence being considered (described briefly immediately before the list of operators). (For clarity, k and l are the number of sequences in the tuples `me` and `other`.)
> >
> > **10. Fig3 the semantics of the operation in L2 have not been defined, similarly for Fig2 L7.**
> > Fig3 L2: At this point, `other` is a tuple of sequences, `other=(ot1,,...,otl)`. Line 2 is tuple concatenation, creating `other=(ot1,...,otl,indices)`.
> > Fig2: L7 (and L6) use syntactic sugar which we have implemented in the language:
> > L6 is sugar for `has_earlier=zipmap(num_earlier,lambda n:n>0)`, and
> > L7 is sugar for `masked_hist = zipmap((hist,length,has_earlier),lambda h,l,he:h-(l*he))`.
> >
> > We will clarify all of these in the paper - thank you!

---

> ### Comment · AnonReviewer2 · 2020-11-24
> **Thanks**
>
> Thank you for the clarifications and updates. As I originally mentioned, this is an interesting idea that provides a conceptual framework for thinking about transformers. As such, the presentation and the discussion need to be improved a lot. The RASP formalism also needs to be more precise.
>
> The text in the author response is a good starting point, but the paper needs to be thoroughly reworked to be useful to this community before accepting it. Furthermore, the implications of using RASP as a thinking device, need to be clearly exposed and evaluated, instead of only focusing on the RASP formalism. For example,
>
> * how num heads, num layers affect the ability to reason over some tasks?
> * Which (practical) tasks would require improved architectures beyond standard transformers? What are the limitations of transformers for reasoning that is needed in common tasks? How can we improve/change transformers thanks to the insights of RASP? What about various "efficient" transformer approximations?
> * For what kinds of reasoning does scaling transformers up (e.g. GPT-3++ and variants) is always doomed to fail?
> * How well do RASP primitives map to the operations that transformers _actually_ learn to perform?
> * _etc._
>
> It would greatly help the community for this paper to provide hints or answers to such questions directly in the paper and present a concise thesis that provides a new point of view.
>
> A side-point: I understand that exposing all these in a few pages of an ICLR/ICML/NeurIPS/AAAI paper is hard. The authors could consider submitting to a journal instead.

---

> > ### Author Response · Authors · 2020-11-24
> > **Thank you, and some directions**
> >
> > Thank you for your time and comments.
> >
> > We would like to quickly direct you to parts of the revised paper, which we believe address several of your comments.
> >
> > Most importantly, we would like to address your comment "how well do RASP primitives map to the operations that transformers actually learn to perform?" . Please consider Figure 5, which we have added to our revised paper (page 9). We show side by side how a transformer trained to compute histograms over the input tokens creates the (non-trivial!) attention pattern exactly matching that predicted by RASP. Other examples can be seen in the full experiments section (Appendix B).
> >
> > Some other comments:
> > - *How num heads/layers affects reasoning ability:*
> > Please see in appendix B the addition of an experimental section. We show cases where RASP accurately predicts the minimal number of layers a task requires, with a marked difference in the success of networks with too little layers as opposed to those with enough.
> > - *How can we improve/change transformers thanks to the insight of RASP?*
> > RASP draws attention to the lack of loops in transformers (and so, to the fact that no task *requiring* loops is realizable in transformers). An existing approach to overcome this is that of Universal Transformers [1]. Other tasks may be recognized in the future as abstractions for feed-forward and attention layers are created, as they will be possible to plug directly into the RASP definition. Please note that the goal of this paper is not to suggest new transformer architectures but to allow us to better understand how they work.
> > - *Discussion of efficient transformers:*
> > We show in the original version that efficient architectures would not be able to implement the operation `count_conditioned` that is available to the vanilla transformer (by using `count_conditioned`  in an efficient transformer to arrive at a contradiction)
> >
> > Additionally, the paper has been edited to make the presentation clearer throughout, for which we thank the helpful comments of all reviewers!
> >
> >
> >
> > [1] Universal Transformers - Mostafa Dehghani, Stephan Gouws, Oriol Vinyals, Jakob Uszkoreit, Łukasz Kaiser

---

### Official Review · AnonReviewer4 · 2020-10-28
**Interesting idea without experimental analysis**

**Rating:** 3
**Confidence:** 4

**Review:**

This paper proposes a programming language, RASP, as a computational model for transformer encoders, and discusses how analysis in terms of this language could be used to understand the behavior of transformer models.

The idea of finding a computational model for transformers is interesting, and (as discussed in section 4) could lead to insights in terms of how to build better models.

However, this paper lacks any results or experimental analysis, which makes it difficult to judge the validity or value of the claims presented. Section 4 discusses how recently proposed transformer variants could be understood (post-hoc) in terms of the RASP language. However, in order to justify using the RASP language to reason about transformers, I think it is necessary to demonstrate experimentally that insights from RASP can translate to new empirical findings.

For example, in section 3.1, the paper makes the claim, “For any given RASP program, we can compute the minimal number of layers required to implement it in a transformer, and upper bound the number of heads this implementation requires.” Can this be verified experimentally, by building a synthetic task, and testing performance as the number of heads is varied?

Similarly, section 4.2 provides an analysis of the recently proposed sandwich transformer model. Could similar analysis be used to make claims about novel, untested architecture variants? Could these claims be verified experimentally? Results such as this would be of high value to the ICLR community.

Because of the lack of experiments, I recommend rejection. I think this is an interesting line of work which could prove valuable to the ICLR community if supported by rigorous experimental evidence.

Minor details:
pg 1: “that is requires” -> “that is required”

---

> ### Author Response · Authors · 2020-11-17
> **Response - Reviewer #4**
>
> Thank you for interest and thoughtful suggestions, we will be attempting to incorporate them into our work in the coming days. In particular, we will run some experiments on synthetic tasks. This will measure how well our example RASP-programs reflect the representations eventually learned by a transformer.
>
> To your question on whether the language could be used to make claims about novel, untested architecture variants, and whether these claims could be verified experimentally, we expect the answer will be yes - just as it was relevant for sandwich transformers and just as we hope the analysis will be relevant for efficient transformers. We also expect such analyses will be valuable to the community!

---

> ### Author Response · Authors · 2020-11-24
> **Experiments added**
>
> Thank you again for your comments.
>
> We have recently uploaded a revision with an experiments section. The full section is in appendix B and some key results have been added to the main paper (section 5). We would like to direct you in particular to figure 5 in the main paper, where we see side by side the unmistakable similarity between a RASP selection pattern and the attention distribution of a transformer trained on the same task. Further results, including prediction of number of layers or heads for synthetic tasks, are presented in appendix B.

---

### Official Review · AnonReviewer1 · 2020-10-29
**Official Blind Review #1**

**Rating:** 6
**Confidence:** 3

**Review:**

This paper proposes a computational model for the transformer in the form of a sequence processing programming language named Restricted Access Sequence Processing Language (RASP). The paper shows how RASP can be used to program solutions to tasks that could conceivably be learned by a transformer. The paper argues that considering computational problems and their implementation in the RASP language allows people to "think like a transformer" in the style of symbolic programs. Overall, the paper is well written and easy to follow.

Reasons to accept the paper:
1. The paper provides a novel way of understanding how transformer model works from a programming language perspective.
2. The paper presents solutions in RASP language for simple tasks such as histograms and sorting, and also complicated logic inference task.
3. The paper attempts to build a connection between the operations in RASP language and the computational operations in transformers. This could help analyze the minimally required number of layers and upper-bound number of heads for the transformer to work on a specific task.

Reasons to reject the paper:
1. For general neural network models, which has no explicit attention mechanism but may still be able to learn to reason over various tasks, it is not clear whether the RASP language is still an abstraction. The paper only discusses transformers, but there is no evidence showing that the operations in RASP cannot be completed by a simple multi-layer neural network. In other words, the connection between RASP and transformer may not be unique, and we may use RASP to think like any neural networks.
2. It is not clear whether there exists other forms of programming language that can also "explain" how transformer works, and if so, how the presented one (RASP) is a better abstraction of the transformer model.
3. Although the presented RASP language can help analyze the number of layers and heads required theoretically, there is limited value and insights for improving existing transformers models.

---

> ### Author Response · Authors · 2020-11-17
> **Response - Reviewer #1**
>
> Thank you for your kind review. We provide our responses below.
>
> **1a. It is not clear whether the RASP language is still an abstraction for general NN models (e.g., those with no attention mechanism)**
>
> The RASP language is not intended for all neural models, it is dedicated directly for the transformer model. In particular, it restricts information flow to match that of a transformer. The motivation is that, while some NN architectures have natural abstractions in other models (e.g., RNNs as automata), transformers do not. This paper serves to fill that void specifically.
>
> **1b. There is no evidence showing that the operations in RASP cannot be completed by a simple multi-layer neural network**
>
> The behaviour of a transformer or RASP program cannot be expressed in a simple feed-forward network, because they do not operate on the same type of inputs. In particular, while a simple multi-layer neural network expects only inputs of fixed length, RASP programs -- and indeed transformers -- apply to inputs of variable length (allocating memory accordingly).
>
> **2. It is not clear whether there exists other forms of programming language that can also "explain" how transformer works, and if so, how the presented one (RASP) is a better abstraction of the transformer model.**
>
> RASP captures the restrictions on information flow imposed by transformers when processing a sequence. While we do not claim it is the only language that can be used to reflect this constraint, we do claim that it is a natural and effective language for doing so: writing simple programs in RASP has directly led to us to some surprising revelations in how a transformer may perform different operations. (For example, attempting to count the number of occurrences of a token in RASP uncovers that the attention pattern focusing only on that token is not actually helpful!).
>
> Additionally, we note that RASP is trivially compatible with restricting the element-processing functions `f` and `s` that `zipmap` and `select` receive, such that they can only be appropriate abstractions for the feed-forward and attention operations of a transformer (when these are made).
>
> **3. Although the presented RASP language can help analyze the number of layers and heads required theoretically, there is limited value and insights for improving existing transformers models.**
>
> We already see that RASP has provided insight about various efficient transformer architectures (specifically, it suggests that there are functions which vanilla transformers can compute but linearly-efficient transformers cannot), and that it provides a good explanation for the results of the recently proposed sandwich transformer. We further anticipate that, as more variants on transformers are proposed in the coming years, this language will help us more clearly consider them and evaluate their differences.

---

### Author Response · Authors · 2020-11-22
**Addition of Experiments**

To all the reviewers,

We deeply appreciate your comments on the paper. We believe they have made it much stronger.

We have run several experiments, testing the ability of RASP to correctly predict the behavior of transformers on a number of synthetic tasks. We have added a brief experiments section to the main paper (section 5), and a fuller compilation of the results to the appendix (appendix B). In particular, we invite you to see Figure 5, where we show that the attention pattern of a trained transformer (for histograms, on inputs with beginning-of-sequence tokens) exactly matches the selector of the natural RASP program for the same task.

Thank you!

---

### Decision · Program_Chairs · 2021-01-07
**Final Decision**

**Decision:**

Reject

**Comment:**

The paper presents a computational model for transformer encoders in the form of a programming language (called RASP), shows how to use this language to "program" tasks solvable by transformers, and describes how to use this model to explain known facts about transformer models.

While the reviewers appreciated the novelty of the main idea, the evaluation and the exposition were found to be below the ICLR bar. As a result, the paper cannot be accepted this time around. I urge the authors to prepare a better new version using the feedback from the reviews and discussion.  In particular, the paper would be much stronger with a discussion of how the ideas here can help with improving the transformer model, and whether these ideas generalize to models other than transformers.